# MOVING BEYOND DIFFUSION: HIERARCHY-TO-HIERARCHY AUTOREGRESSION FOR FMRI-TO-IMAGE RECONSTRUCTION

**Xu Zhang**[1,2]**, Ruijie Quan**[1,2]**, Wenguan Wang**[1,2]**, Yi Yang**[1,2†]
[1]The State Key Lab of Brain-Machine Intelligence, Zhejiang University, China
[2]ReLER, CCAI, Zhejiang University, China
https://github.com/XuZhang2/MindHier

## ABSTRACT

Reconstructing visual stimuli from fMRI signals is a central challenge bridging machine learning and neuroscience. Recent diffusion-based methods typically map fMRI activity to a single neural embedding, using it as static guidance throughout the entire generation process. However, this fixed guidance collapses hierarchical neural information and is misaligned with the stage-dependent demands of image reconstruction. In response, we propose MindHier, a coarse-to-fine fMRI-to-image reconstruction framework built on scale-wise autoregressive modeling. MindHier introduces three components: a Hierarchical fMRI Encoder to extract multi-level neural embeddings, a Hierarchy-to-Hierarchy Alignment scheme to enforce layer-wise correspondence with CLIP features, and a Scale-Aware Coarse-to-Fine Neural Guidance strategy to inject these embeddings into autoregression at matching scales. These designs make MindHier an efficient and cognitively aligned alternative to diffusion-based methods by enabling a hierarchical reconstruction process that synthesizes global semantics before refining local details, akin to human visual perception. Extensive experiments on the NSD dataset show that MindHier achieves superior semantic fidelity, $4.67\times$ faster inference, and more deterministic results than the diffusion-based baselines.

## 1 INTRODUCTION

Reconstructing visual stimuli from fMRI signals stands as a fundamental challenge at the intersection of computer vision and cognitive neuroscience. Recent breakthroughs in this field have been largely driven by diffusion-based methods (Esser et al., 2021; Ho et al., 2020; Xue et al., 2024; Xu et al., 2023; Rombach et al., 2022; Lu et al., 2022; Shen et al., 2026a; Fang et al., 2023; Quan et al., 2024; Chen et al., 2025a; Sun et al., 2023; Huo et al., 2024; Chen et al., 2023b; Li et al., 2024; Song et al., 2026; Shen et al., 2026b), which typically align the encoded fMRI embedding with a single neural representation from multimodal models such as CLIP (Radford et al., 2021). Once aligned, this single neural feature serves as a fixed guidance signal throughout the entire diffusion process, progressively transforming random Gaussian noise into a reconstructed image (Fig. 1(a)).

Despite impressive results, relying solely on a single, static neural feature to guide the entire generation pipeline introduces two fundamental limitations. **First**, fMRI signals are inherently hierarchical, in which different brain regions capture coarse semantic content as well as fine-grained perceptual details (Naselaris et al., 2009). However, current methods collapse this rich, multi-level information into a single vector, leading to its underutilization. **Second**, the guidance is temporally invariant, yet generative models operate in a dynamic multi-stage manner: early stages require global semantic constraints, while later stages demand precise structural and textural cues (Rissanen et al., 2023; Dieleman, 2024). Fixed neural guidance is often redundant in the early phase and insufficient in the later phase, creating a mismatch between representation and generation. ***Beyond these limitations***, diffusion models themselves provide limited control points for injecting stage-aware guidance (Zhou et al., 2024), which hinders the effective exploitation of hierarchical brain features.

---

† Corresponding Author: Yi Yang.

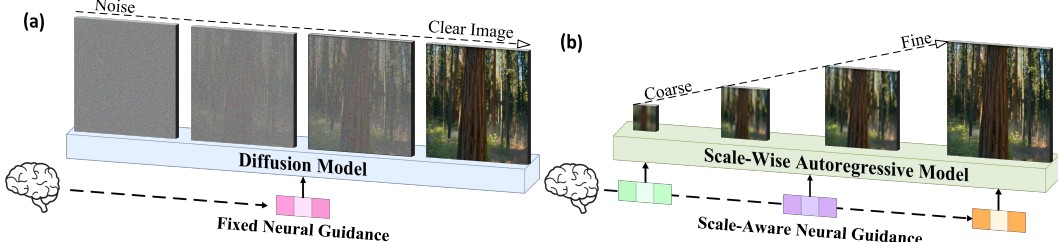

Figure 1: Comparison of fMRI-to-image reconstruction pipelines. **(a)** Prior diffusion-based methods utilize a fixed neural feature to guide the reconstruction. **(b)** In contrast, MindHier employs scale-aware guidance, leveraging hierarchical neural features to first establish a low-resolution overview ("Forest") before progressively refining local details ("Trees") at higher resolutions.

To address these challenges, we propose MindHier, a new coarse-to-fine fMRI-to-image reconstruction framework built upon scale-wise autoregressive modeling (Tian et al., 2024), moving beyond diffusion-based pipelines (Fig. 1(b)). MindHier integrates three key designs. **(i)** *Hierarchical fMRI Encoder*: a unified encoder that transforms fMRI signals into a hierarchy of embeddings, capturing the full spectrum of neural information from global semantics to local details. **(ii)** *Hierarchy-to-Hierarchy Alignment*: a dual-objective training scheme that enforces hierarchy-to-hierarchy alignment between fMRI embeddings and CLIP features, pairing shallow layers with low-level visual features for structural fidelity and deeper layers with high-level features for semantic coherence. **(iii)** *Scale-Aware Coarse-to-Fine Neural Guidance*: a principled strategy that injects the hierarchical embeddings into autoregression at matching scales. Specifically, the high-level semantic embeddings from the fMRI encoder supervise the small-scale (*low-resolution*) stage to establish a semantically coherent global layout, while lower-level embeddings are progressively injected at larger scales (*higher resolution*) to refine structures and enrich textures. Together, these components enable MindHier to leverage the coarse-to-fine nature of visual autoregression, yielding reconstructions that achieve strong semantic fidelity and competitive structural faithfulness compared to diffusion-based baselines. MindHier achieves state-of-the-art high-level metrics on the NSD dataset (Allen et al., 2022), *e.g.*, the highest CLIP score of 96.4% and the lowest SwAV distance of 0.329.

Beyond empirical improvements, MindHier offers several deeper advantages. **First**, its coarse-to-fine reconstruction pipeline naturally echoes the "*Forest before Trees*" principle (Navon, 1977) from cognitive neuroscience, whereby human perception prioritizes global structure before resolving local details. By seeding reconstructions with high-level semantic embeddings and progressively refining fine-grained cues, MindHier translates this perceptual hierarchy into a computational framework. **Second**, the coarse-to-fine autoregressive modeling substantially enhances efficiency: MindHier achieves significant speedups by allocating most computation to low-resolution scales. For example, reconstructing a high-fidelity visual stimulus takes just 2.64 seconds per image, which is 4.67× faster than MindEye2 (Scotti et al., 2024). **Third**, MindHier produces more stable and consistent reconstructions. Unlike diffusion-based pipelines that are inherently stochastic due to random noise initialization, MindHier anchors its generation process directly to fMRI-derived embeddings, initializing with high-level embeddings and conditioning subsequent scales on lower-level ones.

In a nutshell, our **contributions** are three-fold:

- We identify a fundamental mismatch between the fixed neural guidance and the dynamic nature of image reconstruction. To address this, we introduce MindHier, a coarse-to-fine autoregressive framework that dynamically tailors neural guidance to the specific demands of each generation stage, from establishing a global semantic layout to rendering fine-grained details.
- We propose Hierarchy-to-Hierarchy Alignment to disentangle the fMRI signal into hierarchical neural features, and Scale-Aware Guidance to strategically inject these features into the corresponding scales of the autoregressive generator for high-fidelity image reconstruction, resolving the structural mismatch between the neural representation and image reconstruction.
- MindHier establishes a critical balance among high semantic fidelity, deterministic stability, and inference speed. These advantages advance fMRI-to-image reconstruction beyond offline analysis, facilitating the future realization of real-time brain-computer interfaces.

## 2 RELATED WORK

**fMRI-to-Image Reconstruction.** The reconstruction of visual stimuli from brain activity is a pivotal challenge at the intersection of cognitive neuroscience and computer vision. Early works (Kay et al., 2008; Miyawaki et al., 2008; Nishimoto et al., 2011) correlate fMRI signals with hand-crafted image features, while subsequent advances (Horikawa & Kamitani, 2017) improve this brain-visual association by using sparse linear regression to predict features from early CNN layers. The advent of Variational Autoencoders (Kingma & Welling, 2013) and Generative Adversarial Networks (Goodfellow et al., 2014) enables more direct pixel-level reconstruction (Seeliger et al., 2018; Ren et al., 2021; Ozcelik et al., 2022). However, these methods often yield ambiguous images that lack clear semantics. To bridge this semantic gap, researchers have made notable progress by mapping brain signals to the latent space of models like IC-GAN (Perarnau et al., 2016) and StyleGAN (Karras et al., 2019). More recently, the field has been propelled by Latent Diffusion Models (Esser et al., 2021) and multi-modal models like CLIP (Radford et al., 2021). These state-of-the-art methods (Xia et al., 2024a; Ozcelik & VanRullen, 2023; Zeng et al., 2024; Gong et al., 2025; Scotti et al., 2023; 2024; Quan et al., 2024; Li et al., 2025b; Wang et al., 2024) map fMRI signals to the shared embedding space of CLIP to guide the reconstruction. However, these methods suffer from a mismatch between the fixed guidance and the dynamic generation process. In light of this, we propose a distinct framework that adopts a coarse-to-fine generation process with scale-aware guidance, inherently preserving semantic consistency and structural faithfulness to visual stimuli.

**Visual Autoregressive Modeling.** Autoregressive (AR) models have become a cornerstone of generative modeling. Early visual AR approaches (Chen et al., 2020; Ramesh et al., 2021; Bai et al., 2024) adapt the "next-token prediction" paradigm from language models. To do so, they employ discrete tokenizers (Van Den Oord et al., 2017) to quantize images into a 1D sequence using a raster-scan (left to right, top to bottom). However, this process imposes an unnatural 1D bias onto 2D images, often leading to suboptimal performance. A significant shift arrived with Visual Autoregressive Modeling (VAR) (Tian et al., 2024), which introduces a novel "next-scale prediction" paradigm. This paradigm transforms an image into a pyramid of 2D token maps at different scales (resolutions) using a multi-scale residual VQ-VAE tokenizer. Each token map is generated progressively, conditioned on the previously generated ones. Such a paradigm better preserves spatial structure and dramatically improves efficiency without sacrificing quality. Spurred by this innovation, recent works have applied this technique to diverse fields, including depth estimation (Wang et al., 2025), 3D generation (Zhang et al., 2024; Chen et al., 2025b), and text-to-image generation (Ma et al., 2024; Voronov et al., 2024; Han et al., 2025). Despite its proven power, the value of this paradigm for fMRI-to-image reconstruction remains underexplored. Building upon this paradigm, we propose a coarse-to-fine fMRI-to-image reconstruction framework with scale-aware guidance. Our framework reconstructs high-fidelity images and achieves inference speeds up to $4.67\times$ faster.

## 3 METHOD

We introduce MindHier for fMRI-to-image reconstruction, a framework built upon a coarse-to-fine process, which can be broken down into three main components. First, we design a **Hierarchical fMRI Encoder** (§3.1) to transform brain signals into a hierarchical set of neural features. Second, we employ a **Hierarchy-to-Hierarchy Alignment** (§3.2) scheme to train this encoder, ensuring the features capture both fine-grained details and global semantics. Finally, we detail the **Scale-Aware Coarse-to-Fine Neural Guidance** (§3.3), where these learned neural features are used to hierarchically guide a scale-wise autoregressive model during the reconstruction of visual stimuli.

### 3.1 HIERARCHICAL FMRI ENCODER

The central challenge in fMRI-to-image reconstruction lies in bridging the representational mismatch between a static brain signal and the dynamic, multi-stage nature of image reconstruction. A naive approach relying on a single feature would fail to capture this critical perceptual hierarchy. Such a feature provides redundant information for initial coarse synthesis while offering insufficient information for subsequent fine-grained refinement. Our solution is a departure from this approach. We introduce a Hierarchical fMRI Encoder (HFE), an architecture designed not merely to encode the fMRI signal, but to disentangle it into a structured multi-level representation. As illustrated in

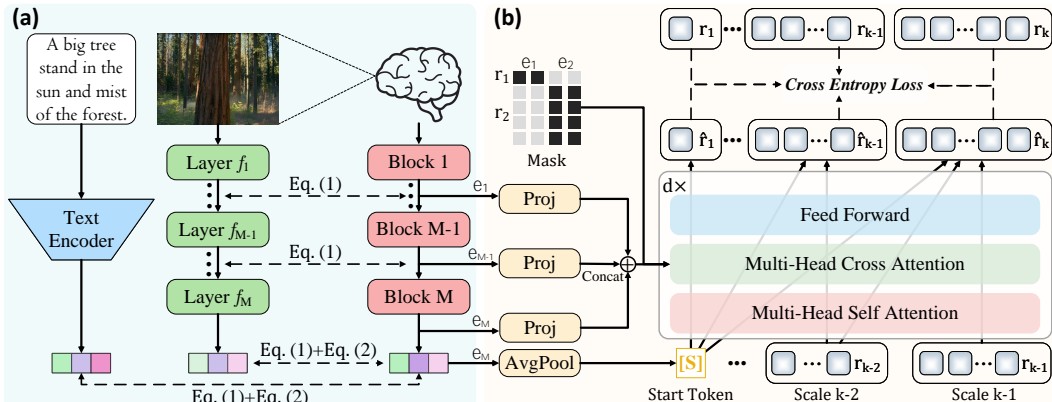

Figure 2: Overview of the two-stage training pipeline of MindHier. **(a)** Stage 1: Hierarchy-to-Hierarchy Alignment. A hierarchical fMRI encoder (composed of $M$ cascaded blocks) is trained to map fMRI signals to a feature hierarchy in CLIP space. This mapping is learned by aligning the encoder's outputs with corresponding intermediate features from a frozen CLIP vision encoder using a cascaded MSE loss ($\mathcal{L}_{\text{MSE}}$ (Eq. 1)). To ensure high-level semantic coherence, the terminal fMRI feature is further aligned within CLIP's shared embedding space via a SoftCLIP loss ($\mathcal{L}_{\text{SoftCLIP}}$ (Eq. 2)). **(b)** Stage 2: Scale-Aware Coarse-to-Fine Neural Guidance. A scale-wise autoregressive model is finetuned to generate images across $K$ scales (coarse-to-fine), conditioned on features from the **fixed fMRI encoder** pretrained in Stage (a). In practice, an attention mask is used to selectively route features via cross attention, which directs the features for the coarse view to attend to the initial scale and the features for finer details to guide subsequent scales. For illustration, a simplified case with block count $M = 2$, fMRI feature dimension $C_{\text{fMRI}} = 2$, and scale count $K = 2$ is shown.

Fig. 2(a), the HFE is realized as a unified stack of $M$ cascaded transformer blocks. Crucially, the outputs from these blocks, $\{\mathbf{e}_1, \ldots, \mathbf{e}_M\}$, are not only intermediate steps but constitute the hierarchical representation itself. This design leverages an established principle from Vision Transformers (ViTs) (Raghu et al., 2021) that ViTs progressively shift from processing local information in initial layers to aggregating purely global information in deeper layers. By explicitly aligning each block's output with a distinct level of a pretrained vision model's feature hierarchy (§3.2), we compel the HFE to learn a structured decomposition of the neural signal. This process culminates in a cascade of features where the terminal output ($\mathbf{e}_M$) encapsulates the global, semantic *"Forest,"* while preceding outputs ($\mathbf{e}_1, \ldots$) preserve the fine-grained *"Trees"*. This tiered representation provides the dynamic, scale-aware guidance (§3.3) that is foundational to our reconstruction framework.

## 3.2 HIERARCHY-TO-HIERARCHY ALIGNMENT

A hierarchical encoder necessitates a hierarchical training objective. To enable the HFE to disentangle the fMRI signal, we introduce a composite loss function that enforces a level-by-level correspondence between the emerging fMRI representations and the feature hierarchy of a frozen CLIP vision encoder. This is achieved through two complementary objectives that are optimized jointly: one ensuring structural alignment and the other enforcing semantic coherence.

**Structural Alignment.** To capture a rich hierarchy of visual percepts, from simple textures to complex object parts, we enforce a direct, structural alignment between the intermediate representations of the HFE and the CLIP vision encoder. This is implemented via a cascaded Mean Squared Error (MSE) loss, which promotes a point-to-point correspondence between the feature sets at multiple levels of abstraction. For each of the $M$ transformer blocks in the HFE, we minimize the distance between its output, $\mathbf{e}_m$, and the feature map from a corresponding vision encoder layer, $\mathbf{v}_{g_m}$:

$$\mathcal{L}_{\text{MSE}} = \sum_{m=1}^{M} \|\ell_2(\mathbf{e}_m) - \ell_2(\mathbf{v}_{g_m})\|_2^2, \tag{1}$$

where $g_m$ is a mapping function from an fMRI block to a specific CLIP layer, and both fMRI features and visual features are $\ell_2$ normalized for improving training stability (Yu et al., 2022).

**Semantic-Level Alignment.** While structural alignment provides the necessary perceptual foundation, it lacks a global semantic anchor to ensure the overall reconstruction is contextually correct. Since "Trees" are conceptually part of the "Forest", any initial error will be propagated through the whole reconstruction process. To address this, we supplement structural alignment with a contrastive objective to bolster semantic alignment in CLIP's shared embedding space. Specifically, we align the terminal HFE feature, $\mathbf{e}_M$, representing the most abstract semantics, with both its visual feature $\mathbf{v}_{g_M}$ and its text caption feature $\mathbf{t}$. This is accomplished using a SoftCLIP loss, which provides a robust semantic anchor through both fMRI-to-image and fMRI-to-text alignment:

$$\mathcal{L}_{\text{SoftCLIP}} = -\frac{1}{B} \sum\nolimits_{i=1}^{B} \left( \log \frac{\exp(\mathbf{e}_i \cdot \mathbf{v}_i / \tau)}{\sum_{j=1}^{B} \exp(\mathbf{e}_i \cdot \mathbf{v}_j / \tau)} + \log \frac{\exp(\mathbf{e}_i \cdot \mathbf{t}_i / \tau)}{\sum_{j=1}^{B} \exp(\mathbf{e}_i \cdot \mathbf{t}_j / \tau)} \right), \qquad (2)$$

where $B$ is the batch size, $\tau$ is a temperature hyperparameter. The explicit feature layer subscripts are omitted for brevity. The total training objective for the fMRI encoder is the sum of the above loss terms, which trains the encoder to learn a hierarchical fMRI feature for subsequent reconstruction.

By enforcing a hierarchical correspondence, this dual alignment ensures that information flows from CLIP to the hierarchical fMRI encoder in a principled manner. The deep-to-deep alignment acts as a conceptual "blueprint", establishing the overall layout first. The shallow-to-shallow alignment acts as a fine-grained "renderer", populating this blueprint with the precise textures, shapes, and structural details perceived by the subject. This structured, top-down alignment is key to reconstruction.

### 3.3 SCALE-AWARE COARSE-TO-FINE NEURAL GUIDANCE

Given the hierarchical neural embeddings $E = \{\mathbf{e}_1, \ldots, \mathbf{e}_M\}$, our objective is to devise a generation process that can leverage these structured embeddings. An ideal generative framework should therefore be able to accept distinct guidance signals at different stages of synthesis. To this end, we build upon scale-wise autoregressive modeling (Tian et al., 2024), a method uniquely suited for our task, as its "next-scale prediction" paradigm provides natural and discrete control points for our scale-aware guidance. In this paradigm, the fundamental regression unit is a 2D token map at a specific scale, where an input image $I$ is first mapped by an encoder $\mathcal{E}$ to a continuous representation $f = \mathcal{E}(I)$, and then discretized by a quantizer $\mathcal{Q}$ into $K$ multi-scale token maps $R = \mathcal{Q}(f) = \{r_1, \ldots, r_K\}$. Each token map consists of discrete indices, where the index at a given position $(i, j)$ is determined by mapping its corresponding input vector to the closest entry in a learned codebook $Z \in \mathbb{R}^{N \times C}$:

$$r_k^{(i,j)} = \underset{n \in \{1,\ldots,N\}}{\arg\min} \|f_k^{(i,j)} - \text{lookup}(Z, n)\|_2. \qquad (3)$$

Here, $f_k^{(i,j)}$ represents the input feature for the $k$-th quantization stage (where $f_1 = f$ and subsequent inputs are residuals), $N = 4096$ is the codebook size, and $\text{lookup}(Z, n)$ retrieves the $n$-th vector from the codebook. The standard autoregressive likelihood is then factorized across these scales, where each token map is generated conditional on all previously generated, lower-resolution maps:

$$p(r_1, r_2, \ldots, r_K) = \prod\nolimits_{k=1}^{K} p(r_k \mid r_1, r_2, \ldots, r_{k-1}), \qquad (4)$$

where each token map $r_k \in \{1, \ldots, N\}^{h_k \times w_k}$ is conditioned on the previously generated maps.

Our key innovation lies in how we condition this scale-wise generation process on our learned fMRI embeddings $E = \{\mathbf{e}_1, \ldots, \mathbf{e}_M\}$. We propose a novel, Scale-Aware Coarse-to-Fine Neural Guidance strategy where the generation of each token map $r_k$ at scale $k$ is conditioned not on a single, fixed embedding, but on a scale-specific conditional feature $\mathbf{s}_k$ from our fMRI hierarchy $E$:

$$p(R|E) = \prod\nolimits_{k=1}^{K} p(r_k | r_{<k}, \mathbf{s}_k), \qquad (5)$$

where $r_{<k}$ denotes the previously generated token maps. The guidance $\mathbf{s}_k$ is dynamically selected from $E$ according to two cognitively inspired phases:

- **Seeding "Forest"** ($k = 1$): The initial, low-resolution scale is responsible for establishing the coarse, overall structure of an image. We therefore provide the embedding ($\mathbf{e}_M$), which captures the most abstract, semantic information, as the special Start Token [S] ($\mathbf{s}_1$) to initialize the generation. This ensures the whole reconstruction process begins with a coherent "Forest" foundation.

Table 1: Quantitative performance comparison on the new NSD test set (Allen et al., 2022). The best and second best results are highlighted in **bold** and underlined respectively. The Wall-clock inference time for one image is reported. †: an auxiliary low-level feature is used.

| Reconstruction Methods | Low-Level | | | | High-Level | | | | Inference |
|---|---|---|---|---|---|---|---|---|---|
| | PixCorr ↑ | SSIM ↑ | Alex(2) ↑ (%) | Alex(5) ↑ (%) | Incep ↑ (%) | CLIP ↑ (%) | Eff ↓ | SwAV ↓ | Time ↓ (s) |
| †BrainDiffuser[Sci. Rep.2023] | 0.273 | 0.365 | 94.4 | 96.6 | 91.3 | 90.9 | 0.728 | 0.421 | 4.87 |
| †MindEye1[NeurIPS2023] | 0.319 | 0.360 | 92.8 | 96.9 | 94.6 | 93.3 | 0.648 | 0.377 | 12.29 |
| †NeuroPictor[ECCV2024] | 0.229 | 0.375 | **96.5** | 98.4 | 94.5 | 93.3 | 0.639 | 0.350 | 8.68 |
| †MindEye2[ICML2024] | 0.322 | 0.431 | 96.1 | 98.6 | 95.4 | 93.0 | 0.619 | 0.344 | 12.14 |
| †MindTuner[AAAI2025] | 0.322 | 0.421 | 95.8 | **98.8** | 95.6 | 93.8 | **0.612** | **0.340** | - |
| †MindHier (Ours) | **0.326**$_{\pm0.0072}$ | **0.461**$_{\pm0.0023}$ | 93.1$_{\pm0.27}$ | 98.0$_{\pm0.04}$ | **95.9**$_{\pm0.21}$ | **95.4**$_{\pm0.39}$ | 0.613$_{\pm0.0024}$ | 0.345$_{\pm0.0037}$ | **2.64** |
| Takagi et al.[CVPR2023] | 0.246 | **0.410** | 78.9 | 85.6 | 83.8 | 82.1 | 0.811 | 0.504 | 15.08 |
| MindBridge[CVPR2024] | 0.151 | 0.263 | 87.7 | 95.5 | 92.4 | 94.7 | 0.712 | 0.418 | 15.98 |
| Wills Aligner[AAAI2025] | **0.271** | 0.328 | **95.8** | 98.0 | 94.3 | 94.8 | 0.649 | 0.373 | - |
| MindHier (Ours) | 0.235$_{\pm0.0060}$ | 0.381$_{\pm0.0017}$ | 94.0$_{\pm0.29}$ | **98.4**$_{\pm0.03}$ | **95.9**$_{\pm0.19}$ | **96.4**$_{\pm0.38}$ | **0.606**$_{\pm0.0021}$ | **0.329**$_{\pm0.0034}$ | **2.64** |

- **Refining "Trees"** ($1 < k \leq K$): The subsequent, higher-resolution scales are responsible for further refinement by adding finer details. At these steps, the model is progressively conditioned on intermediate, detail-focused features, $\mathbf{s}_k = \mathbf{e}_{h_k}$, through a multi-head cross-attention operation. The feature index $h_k$ is defined as $h_k = M - \lfloor M(k-1)/K \rfloor$, where $M \leq K$. The design cleverly provides fMRI features from early transformer blocks to guide the later generation process.

In practice, this scale-aware guidance strategy is implemented via a selective attention mask, as illustrated in Fig. 2(b). This mechanism governs the information flow: initial, coarse scales are constrained to attend only to deep, semantic fMRI features, while subsequent, fine-grained scales are directed towards the shallower, detail-oriented ones. To obtain the final image $\hat{I}$, the quantized vectors for each generated token map $\hat{r}_k$ are summed to approximate a continuous feature map, $\hat{f} = \sum_{k=1}^{K} \text{lookup}(Z, \hat{r}_k)$, and a decoder $\mathcal{D}$ then generates the final image $\hat{I} = \mathcal{D}(\hat{f})$. The entire generative model is trained with a standard Cross Entropy loss to predict the ground-truth token maps at each scale. Ultimately, this pipeline operationalizes our core principle: the image is seeded by a coarse representation of the "Forest" and progressively refined with the detailed "Trees", all guided by a representation learned directly from a unified, hierarchical fMRI encoder.

# 4 EXPERIMENTS

## 4.1 EXPERIMENTAL SETUP

**Dataset.** We use the Natural Scenes Dataset (NSD) (Allen et al., 2022), a publicly available fMRI dataset containing the brain responses of 8 human subjects viewing naturalistic stimuli from COCO (Lin et al., 2014). Following prior work (Takagi & Nishimoto, 2023; Scotti et al., 2024), we use these shared $1,000$ images as our test set, averaging the three fMRI repetitions. The training set consists of the remaining $30,000$ non-averaged, single-trial fMRI-image pairs.

**Training.** Our framework is trained in a two-stage process. First, we train a hierarchical fMRI encoder, with a frozen CLIP ViT/L-14 serving as the visual/textual encoder. Second, we freeze the fMRI encoder and train a scale-wise autoregressive model, pretrained by Switti (Voronov et al., 2024). Due to limited space, more implementation details are provided in Appendix E.

**Testing.** Following a recent work (Scotti et al., 2024), we use the recently updated NSD, which includes $1,000$ testing images. The empirical results of MindHier are obtained on this new test set.

**Evaluation Metrics.** The evaluation spans multiple metrics: low-level similarity (PixCorr = pixelwise correlation between ground truth and reconstructions, SSIM = structural similarity index metric (Wang et al., 2004), AlexNet (Krizhevsky et al., 2012)); high-level semantic features (Incep (Szegedy et al., 2015), CLIP (Radford et al., 2021), Eff (Tan et al., 2019), SwAV (Caron et al., 2020)). Most metrics are reported as two-way identification accuracy, where higher scores are better (↑). Two-way identification accuracy refers to the percent correct if the original image embedding is more similar to its paired fMRI embedding than to a randomly selected fMRI embedding. Eff and SwAV, however, measure the average correlation distance, where a lower score is superior (↓). All results are averaged across subjects 1, 2, 5, and 7, who completed all $40$ scanning sessions.

## 4.2 QUANTITATIVE RESULTS

**Performance Comparison.** Table 1 presents the quantitative comparison of reconstructions of MindHier against several fMRI-to-image baselines. Our evaluation includes the strong MindEye series (Scotti et al., 2023; 2024) and other leading methods (Takagi & Nishimoto, 2023; Ozcelik & VanRullen, 2023; Wang et al., 2024; Quan et al., 2024; Bao et al., 2025). The results demonstrate that MindHier achieves SOTA on several critical metrics. Notably, MindHier surpasses all competitors in semantic fidelity, achieving the top scores for InceptionV3 (95.9%), CLIP (96.4%), and Eff/SwAV distance. This superiority indicates the ability to more accurately capture the core semantic content of visual stimuli. With the addition of the low-level feature used in MindEye2, MindHier$^{\dagger}$ achieves comparable low-level metrics while maintaining SOTA performance on high-level metrics.

**Efficiency Comparison.** Addressing the computational overhead of diffusion models is a key motivation for MindHier. We analyze the computational efficiency of MindHier in Table 1 by comparing its inference time against the leading diffusion-based methods. The results reveal that MindHier is substantially efficient, requiring only $2.64$ seconds to generate a $512 \times 512$ image, marking a $4.67 \times$ speed-up over the MindEye2 model. This computational efficiency is attributable to two key designs: (i) The hierarchical fMRI encoder generates all neural features in a single forward pass. (ii) The coarse-to-fine autoregressive model shifts the majority of computation to lower resolutions. This stands in stark contrast to diffusion models, which must perform many iterative denoising steps across the full-resolution space, making our framework fundamentally more efficient.

## 4.3 QUALITATIVE RESULTS

**High-Fidelity Reconstruction.** We qualitatively evaluate visual reconstruction fidelity by comparing MindHier against leading diffusion-based methods in Fig. 3. Our framework demonstrates superior semantic fidelity across a diverse range of categories, evident in its reconstruction of both simple and complex scenes. For isolated subjects like the giraffe and the teddy bear, MindHier accurately preserves defining characteristics and posture. While other methods may capture the general concept, MindHier often reconstructs a more similar pose and shape. For instance, it successfully renders the distinct long neck of the giraffe, a feature less accurately captured by competitors. More impressively, MindHier excels at preserving key structural and chromatic properties, as shown by the reproduction of the yellow fire hydrant and the teddy bear. Moreover, Mind-

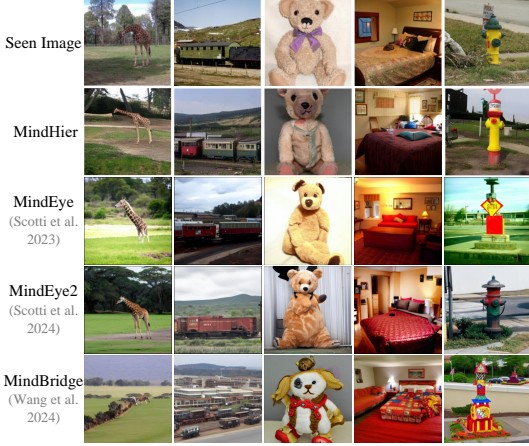

Figure 3: Qualitative comparison of fMRI-to-image reconstructions. The stimuli include a diverse range of classes, from animals and objects to complex indoor and outdoor scenes. All data shown is from Subject 1.

Hier accurately recovers the position, orientation, and trajectory of a running train relative to its environment, while other diffusion-based methods struggle to produce such a coherent result.

**Stable and Consistent Reconstruction.** Beyond reconstruction fidelity, a key advantage of our approach is its stability and consistency across multiple generation trials. This stability stems from our model's deterministic nature. Unlike stochastic diffusion models that begin with a random Gaussian noise seed, our generation process is initiated directly from the encoded fMRI features. As illustrated in Fig. 4, MindHier consistently produces virtually similar reconstructions of the teddy bear across four trials, reliably capturing its core features, such as its shape and posture. In stark contrast, a leading diffusion-based method like MindBridge ex-

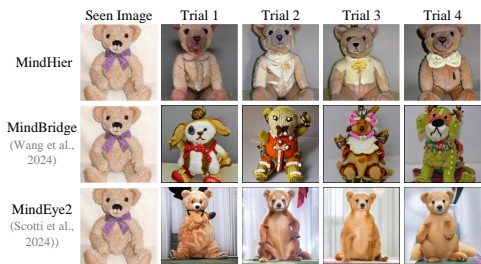

Figure 4: Comparison of reconstruction consistency across four trials. MindHier demonstrates more stable and consistent outputs, in contrast to the stochastic diffusion models.

hibits significant variability; its generations, while semantically related to the "stuffed animal" concept, vary wildly in appearance and color across trials. This high level of consistency underscores the robustness of our method, ensuring that the generated image is a faithful and repeatable decoding of the specific fMRI signal, rather than one of many possible interpretations.

**Faithful Reconstruction.** To further evaluate the ability of MindHier to faithfully reconstruct visual stimuli with high semantic and spatial accuracy, we perform a brain grounding task across a broad range of objects. Following the UMBRAE pipeline (Xia et al., 2024b), this evaluation involves feeding MindHier's reconstructed images into a pretrained Shikra model (Chen et al., 2023a), which then performs localization based on 'known' object labels from the original image's COCO annotation. As illustrated in Fig. 5, the results demonstrate that MindHier consistently generates outputs that are both semantically coherent and spatially accurate. The model successfully localizes not only salient, large-scale objects (*e.g.*, airplane and giraffe) but also smaller, contextual items within a larger scene (*e.g.*, bottle and mouse). Notably, the model successfully reconstructs objects that are partially occluded

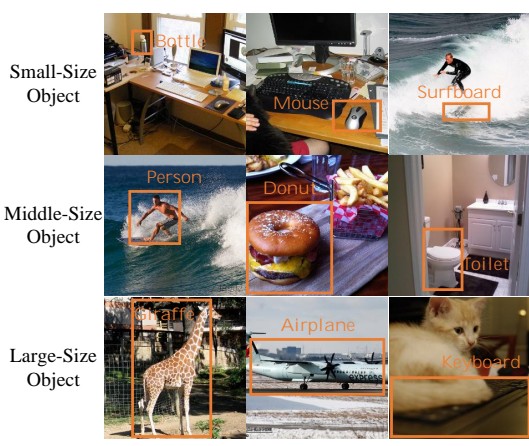

Figure 5: Qualitative results of brain grounding. For visualization, a bounding box generated from each reconstruction is overlaid on the **original visual stimulus**, highlighting the precise localization of the target.

in the original stimulus (*e.g.*, surfboard and keyboard). These results affirm that MindHier's reconstructions are highly faithful to the original stimuli in both structure and spatial arrangement, showcasing a superior capacity to recover critical visual information from brain signals.

## 4.4 DIAGNOSTIC EXPERIMENT

To deconstruct our framework and validate its principal components, we perform three targeted diagnostic experiments. Our analysis is designed to quantify the efficacy of three pivotal design choices: (i) the impact of hierarchical features, (ii) the specific mapping strategy between fMRI encoder blocks and layers of the frozen CLIP vision encoder, and (iii) the role of the scale-aware guidance strategy. To ensure a fair and controlled comparison, all diagnostic experiments are performed on data from Subject 1 while keeping other hyperparameters unchanged.

**Impact of Hierarchical Features.** We first investigate the effectiveness of our core idea to obtain hierarchical features for guiding the coarse-to-fine scale-wise autoregressive reconstruction. As shown in Table 2, our full model, which uses "Hierarchical Feature with our full cascaded supervision (*i.e.*, training with Eq. 1+Eq. 2 for each block)" achieves a substantial performance gain compared with the other two baselines. The first baseline, "Single Feature", only leverages the terminal, high-level fMRI feature for guidance, and the second one, "Hierarchical Feature (final supervision)", uses the same hierarchical architecture as MindHier but omits the MSE loss supervision for intermediate blocks. The empirical results in Table 2 suggest that, although "Hierarchical Feature (final supervision)" shows a modest improvement over "Single Feature" (*e.g.*, $95.1\% \rightarrow 95.4\%$ in CLIP score), our full framework is more favored (*e.g.*, $95.4\% \rightarrow 97.2\%$ in CLIP score). These results suggest that (i) a rich, hierarchical representation that captures both semantic and detailed visual cues is necessary, and (ii) explicit, scale-aware neural guidance is crucial to translating the known representational hierarchy of the pretrained CLIP model to our hierarchical fMRI encoder.

**Selection of CLIP Vision Encoder Layers.** We next validate the design of our CLIP layer mapping strategy. The results are summarized in Table 3. Our method adopts CLIP ViT-L/14 as the visual backbone. Based on the proposed mapping ($g_m = 8 + 4m$), which maps the fMRI signal to the $\{12, 16, 20, 24\}$-th layers, we first derive a late-layer mapping variant, $g_m = 16 + 2m$, which aligns fMRI features with visual feature that encode more semantic information from the deeper layers of the vision encoder (the $\{18, 20, 22, 24\}$-th layers). We then provide an alternative, $g_m = 6m$, an earlier-layer mapping to the $\{6, 12, 18, 24\}$-th layers. We observe consistent trade-offs between low-level and high-level fidelity. The early-layer mapping yields marginal improvements in low-level metrics

Table 2: Comparison of different fMRI encoder designs using fMRI data from Subject 1.

| Neural Representation | PixCorr ↑ | SSIM ↑ | Alex(2) ↑ | Alex(5) ↑ | Incep ↑ | CLIP ↑ | Eff ↓ | SwAV ↓ |
|---|---|---|---|---|---|---|---|---|
| Single Feature | 0.188 | 0.358 | 89.2% | 96.6% | 95.7% | 95.1% | 0.610 | 0.346 |
| Hierarchical Feature (final supv.) | 0.201 | 0.363 | 90.8% | 97.0% | 95.9% | 95.4% | 0.608 | 0.339 |
| Hierarchical Feature (full supv.) | **0.273** | **0.394** | **97.0%** | **99.3%** | **96.7%** | **97.2%** | **0.598** | **0.321** |

Table 3: Comparison of different CLIP layer mapping mechanisms using fMRI data from Subject 1.

| CLIP VIT-L/14 Layer Mapping | | PixCorr ↑ | SSIM ↑ | Alex(2) ↑ | Alex(5) ↑ | Incep ↑ | CLIP ↑ | Eff ↓ | SwAV ↓ |
|---|---|---|---|---|---|---|---|---|---|
| $g_m = 16 + 2m$ | {18, 20, 22, 24} | 0.226 | 0.377 | 92.3% | 98.0% | 95.7% | 95.4% | 0.609 | 0.335 |
| $g_m = 6m$ | { 6, 12, 18, 24 } | **0.283** | **0.399** | **97.6%** | **99.4%** | 96.3% | 94.8% | 0.612 | 0.325 |
| $g_m = 8 + 4m$ | {12, 16, 20, 24} | 0.273 | 0.394 | 97.0% | 99.3% | **96.7%** | **97.2%** | **0.598** | **0.321** |

(*e.g.*, 0.283 vs. 0.273 in PixCorr), while the late-layer mapping degrades overall performance. This indicates that (i) aligning with earlier layers provides more foundational visual features at the cost of high-level semantic accuracy, and (ii) aligning with excessively late layers results in features that are too semantically similar and lack the distinctiveness needed for fine-grained reconstruction. In contrast, our balanced mapping achieves strong high-level similarity (*i.e.*, 97.2% in CLIP) while maintaining excellent low-level similarity (*i.e.*, 99.3% in Alex(5)). This reveals the effectiveness of correlating fMRI signals with a balanced range of intermediate-to-final CLIP layers.

**The Role of the Scale-Aware Guidance.** Finally, we validate the design of our scale-aware neural guidance by conducting an experiment that inverts the information flow. In this variant, the generation is seeded with the first fMRI feature ($e_1$) and progressively guided by coarser semantic features ($e_2, \ldots, e_M$). As summarized in

Table 4: Ablation study on the scale-aware neural guidance. The proposed coarse-to-fine strategy is compared against an inverted variant on Subject 1.

| Method | Incep ↑ | CLIP ↑ | Eff ↓ | SwAV ↓ |
|---|---|---|---|---|
| MindHier (Coarse-to-Fine) | **96.7%** | **97.2%** | **0.598** | **0.321** |
| Inverted (Fine-to-Coarse) | 96.2% | 96.1% | 0.606 | 0.330 |

Table 4, this inversion leads to a clear degradation in high-level metrics, *e.g.*, the CLIP score falls from 97.2% to 96.1%, and the SwAV distance worsens from 0.321 to 0.330. The performance gap confirms that a principled coarse-to-fine information flow is critical for achieving high-fidelity results, where global context precedes fine details. At the same time, the model's resilience is noteworthy; the fact that performance degrades but does not collapse entirely highlights the inherent robustness of the scale-wise autoregressive architecture. We attribute this to the flexibility of the transformer's attention mechanism, which can partially compensate for the suboptimal guidance. Therefore, while the MindHier framework is remarkably robust, our results confirm that its optimal performance is unlocked through the cognitively inspired scale-aware neural guidance strategy.

## 5 CONCLUSION

In this work, we identify a primary obstacle to high-quality fMRI-to-image reconstruction: the mismatch between fixed guidance and the dynamic generation process in prevailing diffusion-based methods. To address this, we introduce MindHier, a framework that reframes this task as a coarse-to-fine, scale-wise fMRI-guided autoregressive problem. MindHier maps fMRI signals to a hierarchy of neural features that are strategically used to guide a reconstruction process, first establishing global semantics at small scales before progressively refining finer details at larger scales. Empirically, this cognitively aligned framework yields reconstructions with superior semantic consistency and structural faithfulness. Furthermore, MindHier achieves a significant 4.67× acceleration in inference speed over leading diffusion-based methods such as MindEye2. Looking forward, we plan to extend MindHier to cross-subject fMRI-to-image reconstruction via multi-subject pretraining and fMRI-to-video reconstruction by incorporating temporal dynamics. It also comes with new challenges and avenues for future research, in particular enhancing the generation of faithful textures, improving the fidelity of facial features, and exploring resource-efficient schemes. Given the rapid evolution in computer vision and cognitive neuroscience over the past few years, we expect more innovations towards these promising directions.

## 6 ACKNOWLEDGMENTS

This research is supported by "Pioneer" and "Leading Goose" R&D Program of Zhejiang (No. 2024C01161), Fundamental Research Funds for the Central Universities (226-2025-00057), and the State Key Laboratory of Brain Cognition and Brain-inspired Intelligence Technology (No. SKLBI-K2025004).

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

## A SUMMARY OF THE APPENDIX

We provide additional details in this supplementary material, which are organized as follows:

- §B elaborates on the data and code availability.
- §C presents more experimental results.
- §D offers more qualitative results.
- §E introduces more implementation details.
- §F provides a deeper analysis of inference time.
- §G details the visual question answering evaluation.
- §H displays expanded subject-specific visualizations.
- §I shows quantitative results on a new Thing-fMRI dataset.
- §J discloses the use of Large Language Model in this work.

## B DATA & CODE AVAILABILITY

**Data.** This work is conducted on the Natural Scenes Dataset (NSD), a large-scale, publicly available fMRI dataset. NSD contains high-resolution, whole-brain fMRI recordings from eight subjects who viewed thousands of colored nature scenes over 30 to 40 scan sessions. The full dataset is accessible upon completion of the NSD Data Access Agreement. Following Scotti et al., 2023; 2024, we use preprocessed flattened fMRI voxels in 1.8-mm native volume space. The data is filtered using the unique "nsdgeneral" mask in NSD, which isolates a subset of voxels that exhibit the strongest response to visual stimuli within the posterior cortex.

**Code.** The supplementary material provides the source code for the core components (fMRI encoder and scale-wise neural guidance) of our proposed method. The complete codebase for both training and inference will be made publicly available upon acceptance. We hope that our findings will encourage more innovations in this research field.

**Pseudo Code.** To guarantee reproducibility, we provide the pseudo codes in PyTorch style for MindHier's key modules, which are given in Algorithm 1 and Algorithm 2.

---

**Algorithm 1** Pseudo-code for Hierarchical fMRI Encoder of MindHier in a PyTorch-like style.

---

```python
# Models:
# fMRI_encoder: The fMRI encoder with M blocks.
# visual_encoder: Frozen CLIP ViT/L-14 visual encoder.
# text_encoder: Frozen CLIP ViT/L-14 text encoder.
def train_fmri_encoder(fmri_signals, images, text_captions):
    # 1. Extract features from CLIP encoders.
    multi_tier_visual_feats = visual_encoder(images, output_hidden_states=True)
    text_feats = text_encoder(text_captions)
    # 2. Map fMRI signals to a hierarchy of predicted features.
    fmri_feats_hierarchy = fMRI_encoder(fmri_signals) # M features
    # 3. Compute alignment losses between predicted and ground-truth features.
    loss_mse = 0
    for m in range(M):
        e_m = l2_normalize(fmri_feats_hierarchy[m])
        v_gm = l2_normalize(multi_tier_visual_feats[m])
        loss_mse += MSELoss(e_\text{M}, v_gm) # Enforces structural correspondence (Eq. 1)
    # Align final fMRI feature with both visual and text features in shared space.
    e_M = fmri_feats_hierarchy[-1]
    v_gM = multi_tier_visual_feats[-1]
    t = text_feats
    loss_softclip = SoftCLIPLoss(e_\text{M}, v_gM) + SoftCLIPLoss(e_\text{M}, t) #
        Enforces semantic correspondence (Eq. 2)
    # 4. Combine losses and update the fMRI encoder.
    total_loss = loss_mse + loss_softclip
    total_loss.backward()
    optimizer.step()
    return total_loss
```

---

## C MORE EXPERIMENTS

We conduct two additional experiments to demonstrate the advantages of the MindHier framework: strong per-subject performance and its resource-free nature.

**Algorithm 2** Pseudo-code for the Scale-Wise Neural Guidance for Image Reconstruction of Mind-Hier in a PyTorch-like style.

```python
# Models:
# fMRI_encoder: The trained fMRI encoder.
# ar_transformer: The scale-wise autoregressive transformer.
# tokenizer_decoder: The decoder part of the multi-scale VQ-VAE.
def reconstruct_image(fmri_signal):
    # Extract multi-tier features from the fMRI signal using the trained encoder.
    fmri_features = fMRI_encoder(fmri_signal) # Batch of M feature tensors
    # --- Stage 1: Seeding the "Forest"
    start_token = mean(fmri_features[-1])
    # Generate the initial, low-resolution token.
    r1_hat = ar_transformer(S=start_token)
    generated_token_maps = [r1_hat]
    # --- Stage 2: Rendering the "Trees" (Progressive, detailed refinement)
    # K is the total number of scales for the autoregressive model.
    for k in range(1, K):
        # Select the appropriate fMRI feature to guide the current scale k+1.
        feature_idx = M - floor(M * k / K)
        scale_cond = fmri_features[feature_idx]
        # Predict the next token map conditioned on previous maps and the fMRI feature.
        rk_hat = ar_transformer(prefix=generated_token_maps, condition=scale_cond)
        generated_token_maps.append(rk_hat)
    # --- Final Image Decoding ---
    # lookup retrieves vectors from the codebook Z.
    f_hat = sum(lookup(Z, r_hat) for r_hat in generated_token_maps)
    reconstructed_image = tokenizer_decoder(f_hat)
    return reconstructed_image
```

**Single-Subject Evaluations.** Table S1 shows exhaustive evaluation metrics computed for each subject. The empirical results demonstrate the model's strong and consistent performance across the cohort. While a non-trivial inter-subject variance is present, which we attribute to the inherent heterogeneity in the underlying data, the model's efficacy is not compromised. Crucially, the reconstruction performance for every subject significantly surpasses established baselines. This indicates that our model exhibits strong robustness and generalization capabilities across diverse individuals, a critical property for real-world deployment.

Table S1: Quantitative results of each subject.

|     |         | S1 | S2 | S5 | S7 |
|-----|---------|-------|-------|-------|-------|
| Low | PixCorr ↑ | **0.273** | 0.237 | 0.218 | 0.213 |
|     | SSIM ↑ | **0.394** | 0.383 | 0.378 | 0.372 |
|     | Alex(2) ↑ | **97.0%** | 95.1% | 91.4% | 92.3% |
|     | Alex(5) ↑ | **99.3%** | 98.8% | 97.8% | 97.6% |
| High | Incep ↑ | 96.7% | 96.1% | **97.0%** | 94.0% |
|     | CLIP ↑ | 97.2% | 96.0% | **97.4%** | 95.0% |
|     | Eff ↓ | 0.598 | 0.611 | **0.576** | 0.640 |
|     | SwAV ↓ | 0.321 | 0.328 | **0.316** | 0.352 |

**Finetuning with One Session of Data.** Table S2 shows that the coarse-to-fine scale-wise autoregressive modeling is resource-efficient. To assess this, we finetune the scale-wise autoregressive model using a single fMRI session as training data and compare it with the results of MindEye2 (Scotti et al., 2024) training with one session of data. We select the two best-performing subjects (S1, S5) on MindEye2 for comparison and find that our results significantly outperform MindEye2 on multiple metrics (*e.g.*, CLIP and Eff scores).

Table S2: Quantitative results with one session of training data. $E$: the results from MindEye2.

|     |         | S1 | $S1^E$ | S5 | $S5^E$ |
|-----|---------|-------|-------|-------|-------|
| Low | PixCorr ↑ | 0.198 | **0.235** | 0.166 | 0.175 |
|     | SSIM ↑ | 0.382 | **0.428** | 0.365 | 0.405 |
|     | Alex(2) ↑ | 86.9% | **88.0%** | 84.2% | 83.1% |
|     | Alex(5) ↑ | **94.8%** | 93.3% | 93.4% | 91.0% |
| High | Incep ↑ | 93.4% | 83.6% | **94.2%** | 84.3% |
|     | CLIP ↑ | 92.2% | 80.8% | **92.7%** | 82.5% |
|     | Eff ↓ | 0.646 | 0.798 | **0.644** | 0.781 |
|     | SwAV ↓ | **0.350** | 0.459 | 0.359 | 0.444 |

**Preliminary Cross-Subject Experiments.** Table S3 presents quantitative results under a cross-subject generalization protocol, where the model is pretrained on a pooled dataset and finetuned on a held-out target subject. In the data-rich regime (40-hour finetuning), MindHier demonstrates superior transferability, achieving a CLIP score of 95.8% and surpassing the MindEye2 baseline (93.5%). This confirms the

Table S3: Quantitative results for cross-subject generalization.

| Method | Incep ↑ | CLIP ↑ | Eff ↓ | SwAV ↓ |
|--------|---------|--------|-------|--------|
| MindEye2 (40 hour) | **96.1%** | 93.5% | 0.609 | 0.338 |
| MindHier (40 hour) | 95.7% | **95.8%** | **0.607** | **0.335** |
| MindEye2 (1 hour) | 83.5% | 80.7% | 0.798 | 0.459 |
| MindHier (1 hour) | 75.4% | 80.2% | 0.860 | 0.527 |

Table S4: Brain Grounding Accuracy (IoU) comparison under single-subject setup.

| Grounding Methods | Eval Subject | All | | Salient | | Salient Creatures | | Salient Objects | | Inconspicuous | |
|---|---|---|---|---|---|---|---|---|---|---|---|
| | | acc@0.5 | IOU | acc@0.5 | IOU | acc@0.5 | IOU | acc@0.5 | IOU | acc@0.5 | IOU |
| MindEye | S1 | 15.34 | 18.65 | 23.83 | **26.96** | 29.29 | 31.64 | **17.88** | **21.86** | 4.74 | 8.28 |
| UMBRAE-S1 | S1 | 13.72 | 17.56 | 21.52 | 25.14 | 26.00 | 29.06 | 16.64 | 20.88 | 4.00 | 8.08 |
| MindHier | S1 | **15.87** | **18.69** | **24.94** | 26.30 | **33.00** | **32.17** | 16.17 | 19.90 | 4.55 | **8.97** |

architecture's capacity to learn robust, transferable representations across individuals. However, in the few-shot regime (1-hour), we observe a performance dip compared to the baseline, which we attribute to overfitting inherent in applying a large-scale autoregressive model to sparse data without specialized regularization, marking a clear direction for future optimization.

**Quantitative Results for Brain Grounding.** Table S4 assesses the spatial fidelity of reconstructions by comparing Brain Grounding Accuracy against established baselines. We benchmark MindHier against MindEye and the specialized grounding method UMBRAE-S1 under the same single-subject setup. In addition to metrics for all classes denoted 'All', UMBRAE groups the 80 classes of COCO according to their prominence into: 'Salient', being the union of 'Salient Creatures' (people and animals) and 'Salient Objects' (*e.g.*, car, bed, table), and 'Inconspicuous' (*e.g.*, knife and toothbrush). The results indicate that MindHier consistently outperforms baselines across most categories, particularly in localizing salient creatures. This quantitative evidence corroborates our qualitative findings, suggesting that our hierarchical guidance mechanism facilitates more precise object localization and superior spatial faithfulness compared to methods relying on global or static conditioning.

**Single-Shot vs Best-of-N.** Table S5 investigates the trade-off between sampling efficiency and reconstruction quality. While our primary evaluation utilizes a Best-of-N (N=4) selection strategy to ensure rigorous comparison, the single-shot (N=1) performance remains re-

Table S5: Comparison of Single-Shot (N=1) generation versus Best-of-N selection.

| Samples | PixCorr ↑ | SSIM ↑ | CLIP ↑ | SwAV ↓ | Time (s) |
|---|---|---|---|---|---|
| N=1 | 0.234 | 0.380 | 95.0% | 0.333 | 0.92 |
| N=4 | 0.235 | 0.381 | 96.4% | 0.329 | 2.64 |

markably robust, achieving a CLIP score of 95.0% with a sub-second inference time of 0.92s. This indicates that MindHier exhibits high determinism and stability, capable of generating high-fidelity reconstructions without reliance on stochastic over-sampling.

## D MORE QUALITATIVE RESULTS

**Neuroscience Interpretability.** To evaluate the neuroscience interpretability of the Mind-Hier framework, we follow MindDiffusion (Lu et al., 2023) and employ the pycortex library (Gao et al., 2015) for brain activity visualization. We utilize L2-regularized linear regression models to predict fMRI voxel responses from features extracted from different blocks of our proposed fMRI encoder. As mentioned in Appendix B, there exists a partial overlap between voxels selected via the manually defined "**nsdgeneral**" mask and those within anatomically defined visual cortex regions.

To specifically probe the relationship between the model's hierarchy and the brain's visual processing stream, we focus our analysis on this overlapping subset of voxels. Fig. S1 (first row) visualizes the results on a flattened cortical surface, where the early visual cortex is demarcated in red and higher-order visual cortices are in blue. The analysis reveals that different model blocks exhibit functional specialization in predicting brain activity across the visual hierarchy. Specifically, early blocks are more predictive of activity in the early visual cortex, whereas deeper blocks show a correspondence with activity in higher-order visual regions. This functional gradient demonstrates

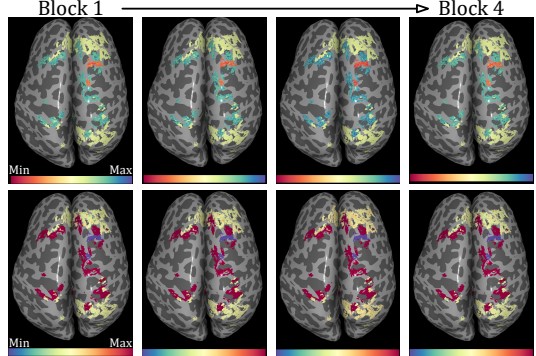

Figure S1: The importance of different regions in decoding fMRI features from each transformer block.

a clear hierarchical alignment between the model's successive layers and the progressive stages of the primate ventral visual stream. Intriguingly, non-visual brain regions (depicted in yellow;

Fig. S1, second row) also exhibit a similar response profile, suggesting that features from intermediate model layers may also reflect the importance of other information in brain activity. Although MindHier does not utilize explicit ROI-based supervision, the model spontaneously learns these functional specializations solely from the data within the "nsdgeneral" mask.

**Illustration of the Coarse-to-Fine Reconstruction Process.** Fig. S2 provides a qualitative illustration of the coarse-to-fine generative process integral to the MindHier framework. This procedure is designed in principled alignment with the "Forest before Trees" tenet from cognitive neuroscience, which posits that human perception begins with a global, coarse overview before proceeding to finer details. As depicted, the reconstruction for each stimulus commences not from random noise, but from an initial low-resolution base that establishes the overall semantic context and foundational structure. Through a scale-wise autoregressive modeling paradigm, the framework progressively refines this initial representation by rendering details at successively higher-resolution scales. This refinement is

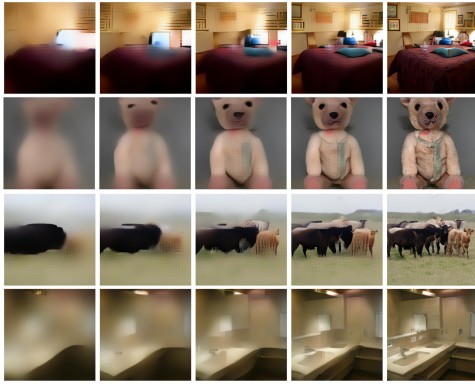

Figure S2: Illustration of the coarse-to-fine reconstruction process of MindHier.

fundamentally distinct from the current dominant diffusion-based methods that typically recover an image from unstructured Gaussian noise through an iterative denoising process.

**Comparison of Reconstructions Across Subjects.** Fig. S3 provides a qualitative comparison of image reconstructions generated from the fMRI data of four different subjects (S1, S2, S5, and S7). The results demonstrate a high degree of consistency in reconstruction quality across these individuals, indicating that the model robustly captures the salient semantic and structural features of the visual input. For instance, across all four subjects, the model successfully reconstructs the defining characteristics of diverse objects, such as the octagonal shape of a stop sign, the stripes of a zebra, and the form of an airplane. Although minor inter-subject variability is present in the finer details, which is expected given the inherent differences in neural activity patterns, the core semantic content is consistently and faithfully preserved. This

Figure S3: Qualitative comparison of reconstructions from four different subjects (S1, S2, S5, S7).

suggests that the proposed framework effectively handles different individuals, decoding shared visual information from distinct brain signals.

**Speed vs. Quality.** To verify that our reported speedup is a result of fundamental architectural efficiency rather than a simple reduction in generation steps, we conduct a controlled ablation study. In our primary evaluation, we utilize the official recommended settings for MindEye2 (38 denoising steps) to ensure it performs at its optimal reconstruction quality. Here, we forcibly restrict the diffusion baseline to 10 denoising steps to match the 10 autoregressive scales used by MindHier, thereby aligning the number of generation passes. As illustrated in Fig. S4, this constraint exposes a critical limitation in diffusion-based decoding. While MindEye2 produces high-quality results at 38 steps,

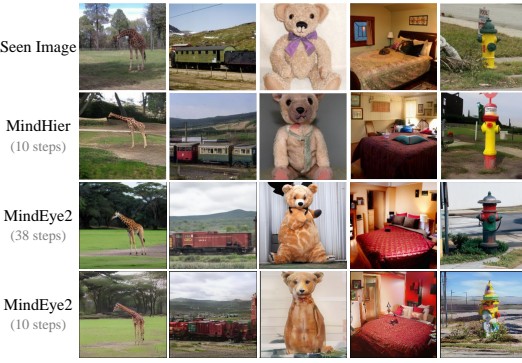

Figure S4: Qualitative comparisons within standard and equalized generation steps.

reducing the step count causes a severe degradation in semantic fidelity. For instance, the fire hydrant (Column 5) loses its structural integrity in the 10-step MindEye2 output, hallucinating a distorted, clown-like figure rather than a distinct object. Similarly, the teddy bear (Column 3) suffers from a loss of identity, resulting in a generic, less coherent animal form. This comparison demonstrates that MindHier's efficiency is not a trade-off between quality and speed; rather, it represents a more effective utilization of computational steps.

## E    MORE IMPLEMENTATION DETAILS

The MindHier frameworks follow a standard two-stage training pipeline by first training a hierarchical fMRI encoder, and then finetuning scale-wise AR with the frozen fMRI encoder.

For optimization of the hierarchical fMRI encoder, we employ AdamW with betas of $(0.9, 0.999)$ and a mini-batch size of $512$. We adopt a cosine annealing policy with a warmup period to schedule the learning rate, which is initialized to $1 \times 10^{-4}$. The model is trained for a total of $300$ epochs.

Regarding the alignment mechanics, the mapping is established between the fMRI feature maps (shape: batch, seq_len, dim) and the corresponding full patch token outputs from CLIP ViT-L/14. To ensure robust optimization, both sets of features undergo L2-normalization prior to the computation of the Mean Squared Error (MSE) loss. For the loss configuration, the total objective is a weighted sum of the MSE loss and the SoftCLIP loss. To ensure training stability, we assign a higher weight to the MSE term $(2 \times 10^5)$ to balance the loss terms, as it presents a more challenging optimization landscape. The SoftCLIP temperature parameter $\tau$ is set to a fixed value of 0.005.

The architecture of the scale-wise autoregressive model in MindHier incorporates modern designs of Switti (Voronov et al., 2024). The model consists of 30 non-causal transformer layers, each of which is mainly composed of a multi-head self-attention block, a multi-head cross-attention block, and a feed-forward network employing the SwiGLU activation function (Shazeer, 2020). For enhanced training stability and efficiency, the model applies RMSNorm (Zhang & Sennrich, 2019) before and after the attention and FFN blocks. It is also employed for query-key (QK) normalization within the attention mechanism. To accommodate variable input resolutions, the model incorporates RoPE (Su et al., 2024; Heo et al., 2024) as positional encoding. Finally, the fMRI embedding is integrated to condition the model via AdaLN (Xu et al., 2019). Second, we freeze the fMRI encoder and train a scale-wise autoregressive model, pretrained by Switti (Voronov et al., 2024). We use the AdamW ($\beta_1 = 0.9, \beta_2 = 0.95$) for optimizer and initialize the learning rate with $1 \times 10^{-4}$, which is also scheduled via cosine annealing with warmup. Training images are augmented using random crops of $512 \times 512$ resolution. There are 10 progressive scales for the autoregressive model. At this stage, we use a mini-batch size of $80$ and train for $9K$ iterations. During inference, we follow MindBridge (Wang et al., 2024) and MindEye (Scotti et al., 2023) to reconstruct 4 candidates of each sample for selecting the best one based on CLIP similarity.

All inference tasks are conducted on a single NVIDIA RTX 4090 GPU. Unlike prior state-of-the-art methods, MindHier does not require auxiliary retrieval submodules or separate captioning networks, relying solely on the unified encoder and autoregressive generator.

## F    A DEEPER ANALYSIS OF INFERENCE TIME

In this section, we conduct two experiments to analyze the efficiency gains of the MindHier model.

**Component-wise Efficiency Analysis.** To investigate the source of our efficiency gains, we analyze the stage-wise inference latency compared to the diffusion-based baseline, MindEye2. As summarized in Table S6, the total inference process is dominated by the reconstruction phase. MindHier achieves a significant speed up through two structural advantages:

Table S6: Quantitative results for visual question answering evaluation.

| Model | Encoding | Reconstruction | Refinement |
|---|---|---|---|
| MindEye2 | 0.785s | 7.864s | 3.323s |
| MindHier | 0.008s | 2.295s | N/A |

- **Elimination of Refinement Stage.** Unlike diffusion-based methods such as MindEye2, which rely on a separate, computationally expensive refinement stage (3.32s) to enhance image quality, MindHier generates high-fidelity outputs in a single autoregressive pass. This structural difference alone removes a major computational bottleneck, eliminating the refinement time.

Table S7: Stage-wise inference time breakdown across autoregressive scales. The latency correlates with image resolution and the majority of inference time is spent on the final high-resolution scales.

| Stage | 0 (Coarse) | 1 | 2 | 3 | 4 | 5 | 6 | 7 | 8 | 9 (Fine) |
|---|---|---|---|---|---|---|---|---|---|---|
| Resolution | 16 | 32 | 48 | 64 | 96 | 144 | 202 | 288 | 384 | 512 |
| Time (s) | 0.028 | 0.031 | 0.038 | 0.040 | 0.074 | 0.216 | 0.126 | 0.283 | 0.474 | 0.985 |

- **Efficient Encoding and Reconstruction.** Beyond the refinement stage, MindHier demonstrates superior efficiency in the core encoding and reconstruction steps. The encoder operates faster than the baseline (0.008s vs. 0.785s). Furthermore, the reconstruction phase is accelerated (2.295s vs. 7.864s) by employing a discrete, scale-wise paradigm rather than iterative diffusion sampling.

**Scale-wise Latency Breakdown.** We further dissect the reconstruction time across different resolutions in Table S7. The results reveal that the computational cost is non-uniform and heavily back-loaded. The coarse stages (Stages 0–4), which establish the semantic structure of the image, are executed with negligible latency (under 0.1s combined). The majority of the inference time is incurred during the final high-resolution upscaling (Stage 9, 0.985s), where the model resolves fine-grained pixel details. This confirms that while pixel-level generation at $512 \times 512$ resolution remains the most demanding step, the hierarchical approach efficiently offloads the semantic planning to lower, faster resolutions.

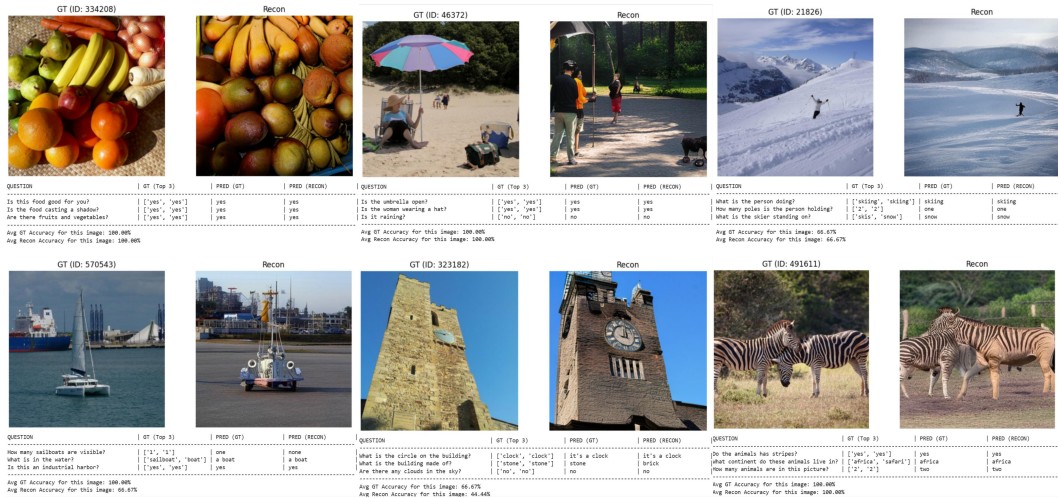

Figure S5: Qualitative results of visual question answering.

# G    VISUAL QUESTION ANSWERING EVALUATION

To assess faithfulness beyond object detection, we extend our evaluation to a Visual Question Answering (VQA) task. Utilizing the pre-trained BLIP-2-OPT-2.7B model, we interrogate the reconstructed images to determine if the semantic content preserved by the brain decoding pipeline is sufficient for a Vision-Language Model (VLM) to interpret correctly.

Table S8: Quantitative results for visual question answering evaluation.

|  | GroundTruth | MindHier | MindEye2 |
|---|---|---|---|
| Accuracy | 58.68% | 43.29% | 41.71% |

**Quantitative Results.** As shown in Table S8, we measure the top-3 accuracy of the VQA model's responses to the reconstructed images against the ground truth answers. The original Ground Truth (GT) images yield a VQA accuracy of 58.68%, establishing the upper bound for the model's reasoning capabilities. MindHier achieves an accuracy of 43.29%, surpassing the previous state-of-the-art MindEye2 (41.71%). This performance gap indicates that MindHier is more effective at faithfully decoding and preserving the distinct semantic features.

**Qualitative Results.** Fig. S5 reveals that MindHier excels at retaining the "gist" and categorical identity of the scene, though fine-grained discrepancies remain:

- **Semantic Retention:** In distinct scenarios, such as the fruit display (ID: 334208) and the zebras (ID: 491611), the reconstructions successfully trigger correct VQA responses regarding object presence ("fruits and vegetables"), attributes ("stripes"), and location ("Africa"). This suggests high fidelity in faithfully decoding dominant subjects.
- **Texture and Material Ambiguity:** While object classes are generally preserved, specific material properties can be altered during reconstruction. For instance, in the tower scene (ID: 323182), the VQA model correctly identifies the "clock," but misidentifies the building material as "brick" rather than "stone" in the reconstruction, pointing to a loss of high-frequency textural information.
- **Fine-Grained Counting Limitations:** The evaluation also highlights challenges with precise counting and small details. In the skiing example (ID: 21826), while the action ("skiing") is correctly classified, the reconstruction fails to preserve the exact number of ski poles held. Similarly, in the harbor scene (ID: 570543), the reconstruction distorts the vessel enough that the model fails to count "one" sailboat, despite correctly identifying the context as a harbor containing a boat.

Overall, while pixel-perfect reconstruction remains a challenge for fMRI decoding, MindHier demonstrates a superior capacity for faithfully reconstructing brain signals into semantically coherent visuals that align closer to the ground truth interpretation than competing methods.

## H  EXPANDED SUBJECT-SPECIFIC VISUALIZATIONS

To provide a comprehensive qualitative assessment of MindHier, we present an expanded gallery of reconstructions of Subj 1, 2, 5, and 7 in Fig. S6-S9. This diverse collection covers a wide spectrum of semantic categories—including wildlife, transportation, food, sports, and indoor environments—demonstrating the model's ability to generalize across varied visual stimuli.

**Semantic Consistency and Object Identity.** The results highlight MindHier's strength in semantic grounding. For distinct, salience-heavy objects, the reconstructions are remarkably accurate.

- **Animals:** In images of animals (*e.g.*, zebras, giraffes, bears), the model successfully captures not only the species but also the pose and context (*e.g.*, a bear in water, a zebra grazing).
- **Transportation:** Rigid structures like airplanes, trains, and buses are reconstructed with correct orientation and scale relative to the background.
- **Food:** Culinary items like pizza and cakes are generated with appetizing fidelity, preserving the general category and color palette, though specific toppings may vary.

**Scene Layout and Atmosphere.** Beyond single objects, the model shows competence in scene understanding. In complex indoor environments (*e.g.*, bedrooms, bathrooms), the spatial arrangement of furniture, such as the placement of a bed relative to a window, is generally preserved. Similarly, the global lighting conditions and color histograms (*e.g.*, the blue hues of a surfing scene vs. the warm tones of a dining room) are consistently matched, indicating effective decoding of low-level visual features alongside high-level semantics.

**Limitations and Failure Cases.** Despite these successes, a closer inspection reveals characteristic failure modes common in fMRI decoding:

- **Fine-Grained Details:** While the "gist" is preserved, specific high-frequency details often hallucinate. For example, while a "pizza" is correctly reconstructed, the exact arrangement of pepperoni slices differs from the ground truth.
- **Text and Symbols:** The model struggles to reconstruct legible text or specific logos (*e.g.*, on signage or vehicles), treating them instead as generic texture patterns.
- **Human Identity:** While the presence and posture of humans are detected, facial features remain generic or blurred, lacking the precision required for identity recognition.

## I  GENERALIZABILITY ON THE THINGS DATASET

To demonstrate the robustness and generalizability of MindHier beyond the NSD benchmark (Allen et al., 2022), we extend our evaluation to the THINGS-fMRI dataset (Hebart et al., 2023). This dataset presents a challenging scenario due to its distinct acquisition protocols compared to NSD.

Table S9: Quantitative results on THINGS-fMRI test set. The best results are highlighted in **bold**.

| Method | PixCorr ↑ | SSIM ↑ | Alex(2) ↑ | Alex(5) ↑ | Incep ↑ | CLIP ↑ | Eff ↓ | SwAV ↓ |
|---|---|---|---|---|---|---|---|---|
| BrainFlora[MM2025] | 0.079 | 0.348 | 0.595 | 62.2% | 60.4% | 57.6% | **0.812** | 0.661 |
| MindHier (Ours) | **0.109** | **0.357** | **0.710** | **81.9%** | **71.9%** | **73.1%** | 0.900 | **0.589** |

**Experimental Setup.** The THINGS-fMRI dataset includes fMRI recordings from three subjects viewing 720 concepts (12 images per concept) during training and 100 concepts (1 image per concept) during testing. To ensure a rigorous and fair comparison with the state-of-the-art method on this benchmark, BrainFlora(Li et al., 2025a), we adopt their exact experimental setting.

Specifically, we employ a joint-subject training strategy where fMRI voxels from all three subjects are padded to a unified dimension (7000). This padding strategy accommodates the varying voxel counts across subjects, allowing our hierarchical encoder to be trained simultaneously on data from all three individuals. We use the same hyperparameters described in the main text (§4.1), adapted only for the input dimension changes required by the THINGS dataset.

**Quantitative Analysis.** Table S9 presents the quantitative comparison between MindHier and the baseline BrainFlora on the THINGS dataset. The results demonstrate that our hierarchical autoregressive approach generalizes effectively to new data distributions and multi-subject settings.

Most notably, MindHier exhibits substantial improvements in high-level semantic retrieval metrics. We observe a **19.7%** increase in AlexNet(5) accuracy ($62.2\% \rightarrow 81.9\%$) and a **15.5%** increase in CLIP identification accuracy ($57.6\% \rightarrow 73.1\%$) compared to the baseline. These gains confirm that our Scale-Aware Coarse-to-Fine Neural Guidance successfully captures and generates the semantic "gist" of diverse concepts found in the THINGS dataset. Furthermore, our method improves low-level structural fidelity, as evidenced by higher PixCorr ($0.079 \rightarrow 0.109$) and SSIM ($0.348 \rightarrow 0.357$) scores. The reduction in SwAV distance ($0.661 \rightarrow 0.589$) further indicates that our generated images are more perceptually aligned with the ground truth in deep feature spaces.

## J  THE USE OF LARGE LANGUAGE MODEL

In the preparation of this manuscript, a Large Language Model (LLM) was employed as a writing assistant to aid in polishing the text and improving clarity. The authors fully developed all concepts, arguments, and conclusions presented. Following the LLM's assistance, every portion of the text was meticulously reviewed, fact-checked, and rewritten by the authors to ensure the final content accurately and entirely reflects our own work and intended meaning.

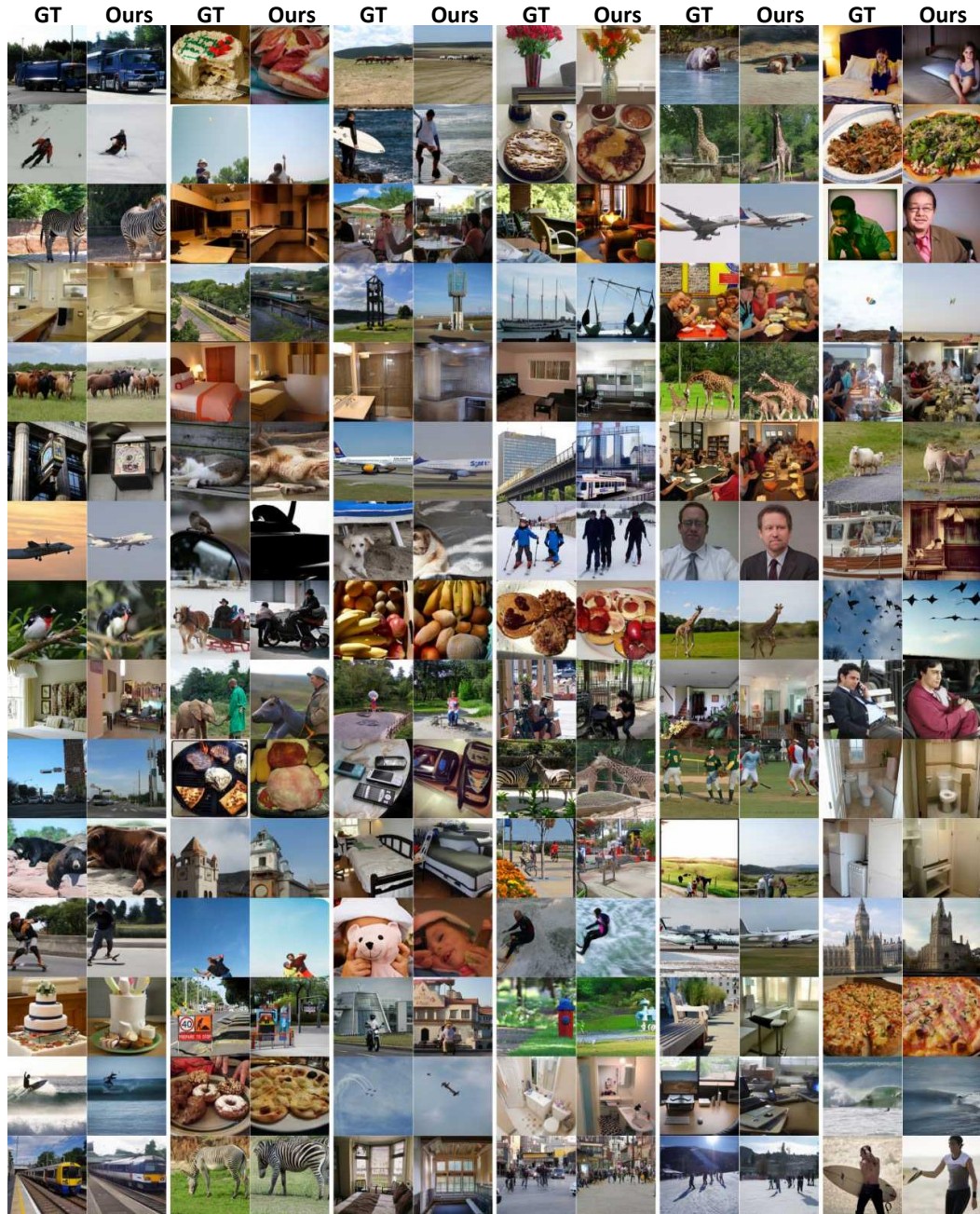

Figure S6: Visualizations of fMRI-to-image reconstruction for Subject-1.

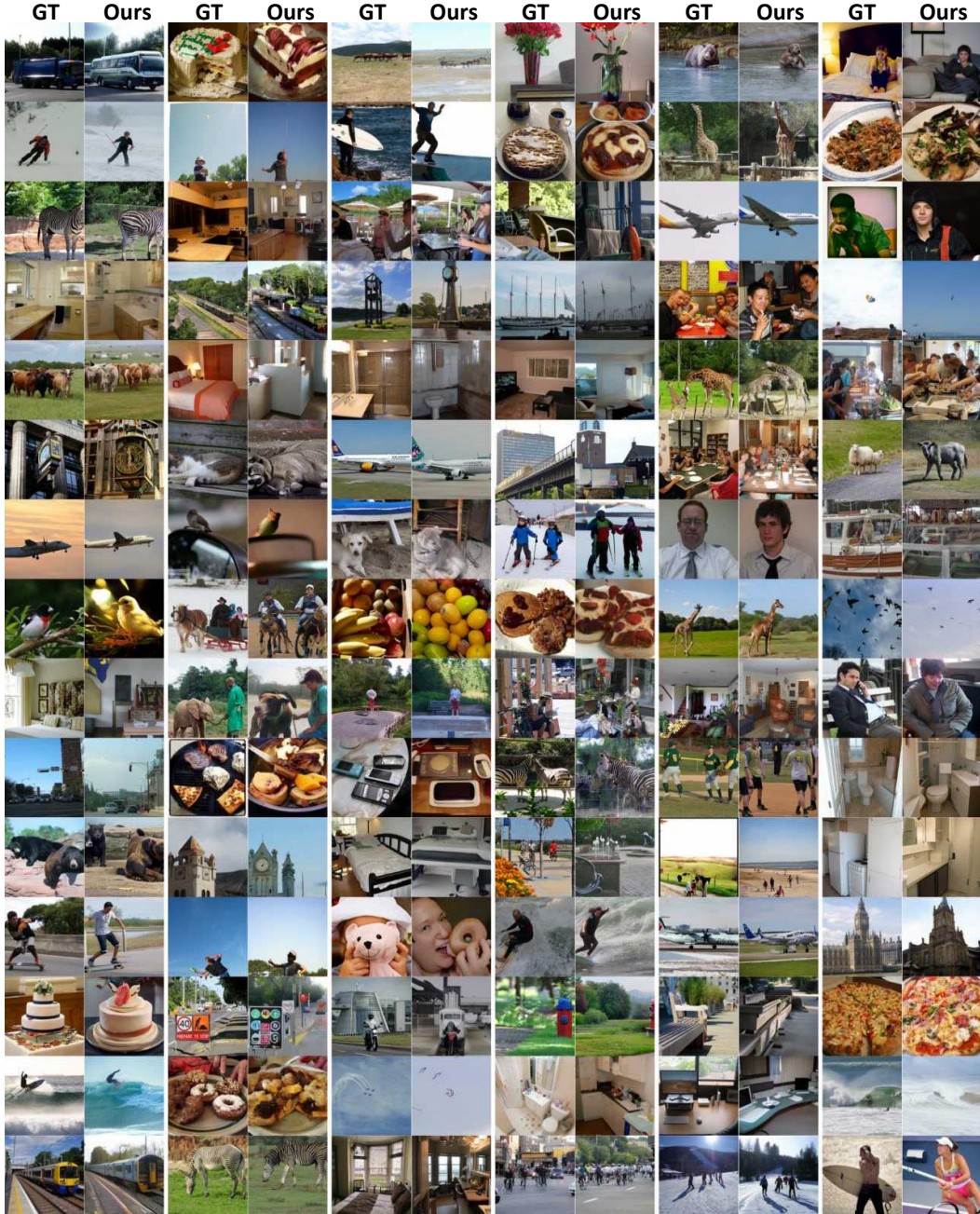

Figure S7: Visualizations of fMRI-to-image reconstruction for Subject-2.

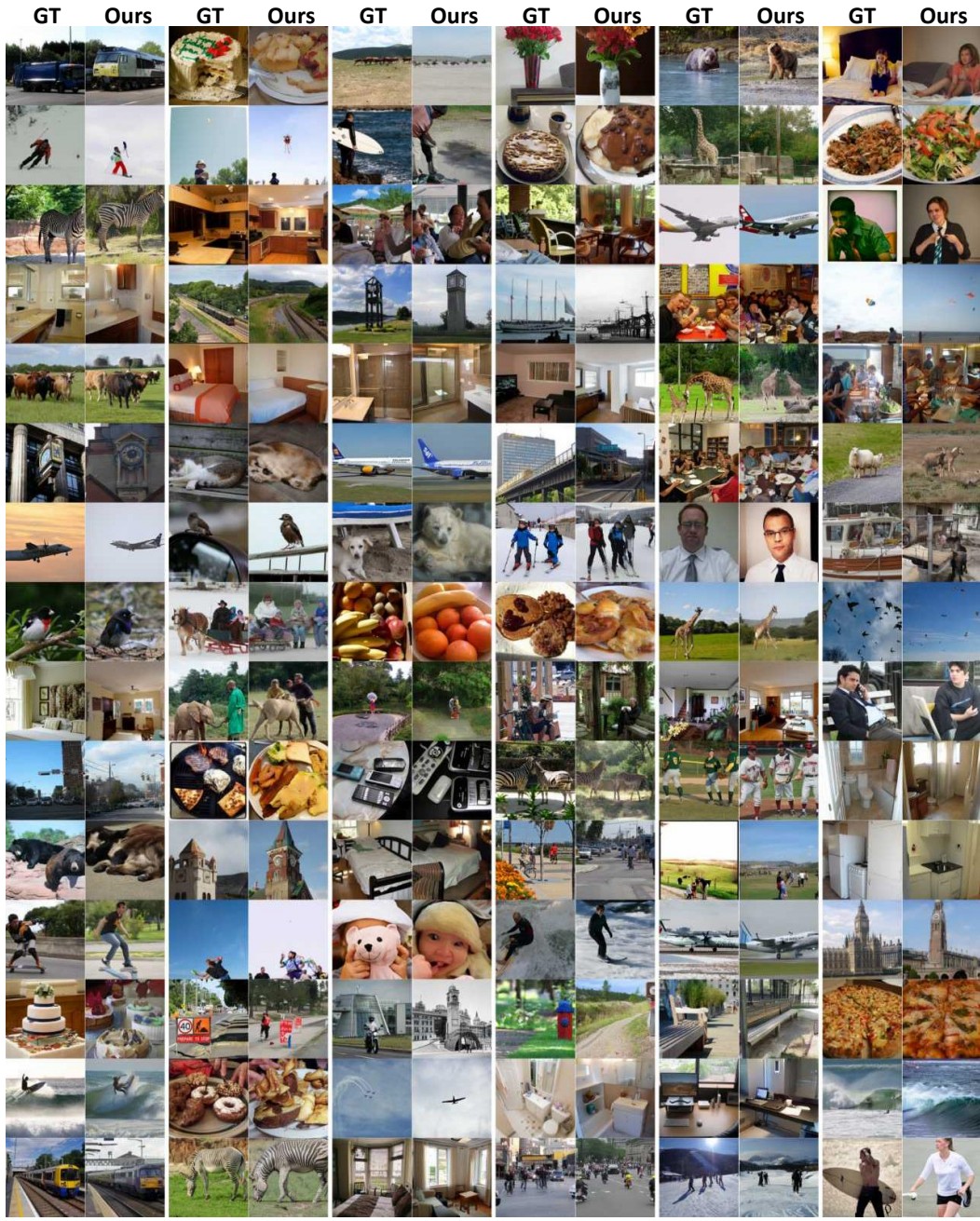

Figure S8: Visualizations of fMRI-to-image reconstruction for Subject-5.

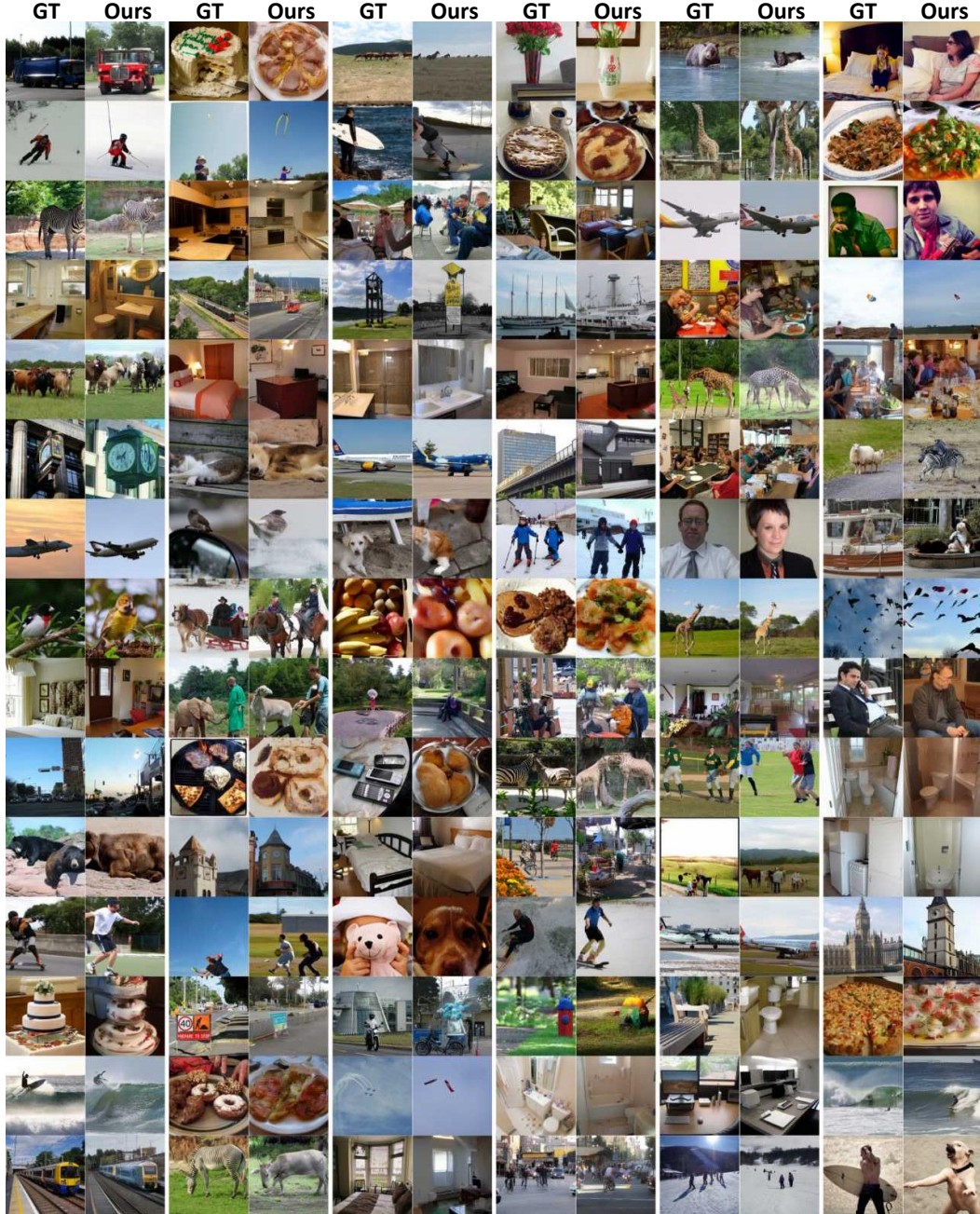

Figure S9: Visualizations of fMRI-to-image reconstruction for Subject-7.

