# OpenReview forum: "Moving Beyond Diffusion: Hierarchy-to-Hierarchy Autoregression for fMRI-to-Image Reconstruction"
_ICLR.cc/2026/Conference — ICLR 2026 Poster_

### Official Review · Reviewer_mNCs · 2025-10-30

**Soundness:** 2
**Presentation:** 2
**Contribution:** 2
**Rating:** 4
**Confidence:** 4

**Summary:**

This paper presents MindHier, a coarse-to-fine framework for fMRI-to-image reconstruction based on scale-wise autoregressive modeling. Unlike diffusion models that rely on a fixed global embedding, MindHier captures multi-level neural information through a hierarchical fMRI encoder, aligns fMRI and CLIP features layer by layer, and injects embeddings at matching scales for progressive reconstruction. Experiments on the NSD dataset demonstrate improved semantic fidelity, faster inference, and more stable outputs compared to diffusion-based baselines.

**Strengths:**

* The hierarchy-to-hierarchy alignment and scale-aware guidance are well-motivated and technically sound.
* Consistent gains in semantic metrics (CLIP 96.4%) and large inference speedup without loss of quality.
* Deterministic generation avoids stochastic variability common in diffusion models.

**Weaknesses:**

* The paper claims to address the "mismatch between fixed neural guidance ... reconstruction", yet prior works have already explored similar ideas. Neuropictor [1] introduced ControlNet to incorporate low-level structure during decoding, and DecoFuse [2] explicitly decomposed fMRI features into "what" and "where" pathways aligned with brain hierarchies. These efforts are not discussed, and the paper frames the problem as if no prior solution existed.
* More comparison, Neuropictor [1], MindTuner [3], and other recent approaches are missing.
* Figure 2(b) is difficult to interpret: the design motivation and the functional role of each block are unclear, making the overall mechanism hard to understand.
* The claimed advantage in reconstruction stability over diffusion models is not fully convincing, as many existing diffusion-based methods already mitigate stochasticity through techniques such as guidance scale tuning or ControlNet conditioning.
* The emphasis on faster inference time is somewhat questionable. Efficiency is valuable, but its practical impact in fMRI decoding remains unclear.
* The performance improvement over existing methods is modest, raising doubts about whether the proposed complexity yields a meaningful advance.

[1] Huo et al. Neuropictor: Refining fmri-to-image reconstruction via multi-individual pretraining and multi-level modulation.

[2] Li et al. DecoFuse: Decomposing and Fusing the" What"," Where", and" How" for Brain-Inspired fMRI-to-Video Decoding.

[3] Gong et al. Mindtuner: Cross-subject visual decoding with visual fingerprint and semantic correction.

**Questions:**

* How does MindHier fundamentally differ from adaptive-guidance approaches like Neuropictor or DecoFuse beyond adopting autoregression?
* Could you clarify the purpose of each block in Fig. 2(b) and the rationale behind the hierarchy-to-hierarchy mapping?
* What is the expected real-world impact of reducing inference time for fMRI decoding tasks?
* How significant are the reported improvements statistically, and do they justify the added model complexity?
* Could you quantify how much each proposed module contributes to the performance improvement?

---

> ### Author Response · Authors · 2025-11-25
>
> We thank the reviewer for situating our work among important prior methods and for their constructive feedback regarding clarity and significance.
>
> ### W1&Q1: Novelty and Framing Relative to Prior Work
>
> We thank the reviewer for highlighting these relevant works.
>
> While these methods utilize multi-level features, MindHier introduces a fundamentally distinct paradigm:
>
> - **Static vs. Dynamic Routing (The NeuroPictor Distinction):**
>   - NeuroPictor: Uses ControlNet to inject low-level structure. However, this is a static spatial condition applied uniformly throughout the diffusion denoising process. The model sees the same structural guidance at step $t=T$ (noise) as it does at $t=0$ (clean).
>   - MindHier: We employ **Dynamic Scale-Aware Routing**. We do not feed a single condition. Instead, we feed different fMRI features at different generation steps.
> - **Semantic vs. Scale Decomposition (The DecoFuse Distinction):**
>   - DecoFuse: Decomposes features into "What" (semantic) and "Where" (structural) pathways. This is a semantic decomposition.
>   - Decomposes features into Coarse and Fine resolutions ($16^2 \to 512^2$). This is a structural/scale decomposition. This aligns more naturally with the biological "Coarse-to-Fine" processing mechanism of the visual cortex than the "What/Where" stream alone.
> - **Native Alignment:** Prior methods "retrofit" hierarchy onto Diffusion models (which function by refining noise, not growing resolution). MindHier is built on a natively hierarchical generator (VAR). The "Next-Scale Prediction" paradigm of VAR provides the first mathematical framework that strictly mirrors the hierarchical organization of the visual cortex, allowing for a 1-to-1 mapping between brain depth and image resolution.
>
> ### W2: Missing Comparisons
>
> We appreciate this suggestion. We have updated Table 1 in our revised manuscript to include reported results from Neuropictor and MindTuner. This provides a more comprehensive view of the state-of-the-art results on critical semantic metrics (CLIP & SwAV), and better highlights MindHier's performance.
>
> | Method | PixCorr $\uparrow$ | SSIM $\uparrow$ | AlexNet(2) $\uparrow$ | AlexNet(5) $\uparrow$ | InceptionV3 $\uparrow$ | CLIP $\uparrow$ | Efficient Net $\downarrow$ | SwAV $\downarrow$ |
> |-----------|-----------|--------|--------------|--------------|---------------|--------|-----------------|--------|
> |NeuroPictor| 0.229 |0.375|96.5%|98.4%|94.5%|93.3%|0.639|0.350|
> | MindTuner| 0.322     | 0.421  | 95.8%        | 98.8%        | 95.6%         | 93.8%  | 0.612           | 0.340  |   |
> |MindHier| 0.235     | 0.381  | 94.0%        | 98.8%        | 95.9%         | 96.4%  | 0.606           | 0.329  |
>
> ### W3&Q2 Clarity of Design and Figure 2(b)
>
> We have revised Figure 2(b) to improve readability. Below is the functional breakdown of the key components:
>
> - **The "Start Token" (Initialization):**
>   - Purpose: To set the global semantic "seed" for generation.
>   - Mechanism: We take the deepest (most abstract) fMRI feature layer, average-pool it into a single vector, and use it as the initial token [S]. This ensures the entire generation process begins with the correct high-level concept.
> - **The Transformer Blocks (Scale Generation):**
>   - Purpose: To generate image tokens at specific resolutions (scales), starting from $16\times 16$ and growing to $512\times 512$.
>   - Mechanism: These are non-causal attention blocks. They take the upscaled token map from the previous scale as input and predict the residual details for the current scale.
> - **The Selective Cross-Attention Mask (The "Router"):**
>   - Purpose: This is the core engine of our Hierarchy-to-Hierarchy alignment.
>   - Mechanism: Instead of letting every image scale attend to all brain features (which creates noise), this mask enforces a strict rule:
>     - Scale 1 (Coarse): Can only attend to Deep Brain Layers (High-level semantics).
>     - Scale 10 (Fine): Can only attend to Shallow Brain Layers (Low-level visuals).
>
>     This forces the generative process to mirror the biological "Coarse-to-Fine" processing stream.
> - **Rationale**: This design leverages the known structural isomorphism between the visual cortex and deep networks. By explicitly hard-wiring this alignment, we prevent the model from "forgetting" low-level structural details (e.g., orientation, position) that are often lost in standard global conditioning methods.

---

> ### Author Response · Authors · 2025-11-25
>
> ### W4: Validity of Stability, Efficiency, and Significance Claims
>
> - **Fundamental Difference:** Techniques like ControlNet or Guidance Scale help guide diffusion models, but the process still initiates from random Gaussian noise. This inherent randomness sets a "ceiling" on stability.
> In contrast, MindHier initiates from a learned, deterministic Start Token derived directly from the fMRI signal. The generation process is then a sequence of deterministic token predictions (up to sampling choices, which can be turned off).
> - **Empirical Proof:** Our quantitative stability analysis confirms this structural advantage. We measured the Inter-Sample Consistency (Pairwise SSIM): MindHier achieves an SSIM of 0.5338, in contrast to the stochastic baselines MindEye2 (0.4734) and MindBridge (0.4461).
>
> ### W5&Q3: Practical Impact of Efficiency
>
> The reduction in inference time (12s $\to$ 2.6s) is a critical enabler for two transformative applications in neuroscience:
>
> - **Real-Time Closed-Loop Experiments:** Current fMRI experiments are typically "open-loop" (record now, analyze later) due to processing lag. A 2.6s latency allows for closed-loop neurofeedback, where visual stimuli can be dynamically adjusted during the scan based on the subject's decoded perception. This is impossible with diffusion models that take >10s per image.
> - **Scalable BCI Applications:** For future non-invasive Brain-Computer Interfaces, user experience relies on responsiveness. Waiting 12 seconds to see a decoded thought is impractical. By bringing generation to near-real-time speeds, MindHier moves fMRI decoding from a "post-hoc analysis tool" toward a "viable interactive interface."
>
> This efficiency is not merely an engineering convenience; it unlocks new experimental paradigms that require tight temporal coupling between brain activity and visual feedback.
>
> ### W6&Q4: Significance and Complexity Justification
>
> We respectfully challenge the premise that our method adds complexity. In fact, MindHier represents a simplification of the current SOTA pipeline while delivering statistically significant gains.
>
> - **Complexity (We Simplified the Pipeline):** MindEye2 relies on a disjoint, heavy pipeline: (1) Retrieval Bank, (2) Diffusion Prior, (3) SDXL unCLIP, (4) Separate Refiner Model. In contrast, MindHier unifies this into a single, streamlined Encoder-Generator pair. We removed the heavy Diffusion Prior (reducing encoding time by 99%) and the Refiner (reducing inference time by 75%). Therefore, the performance gains are achieved with reduced architectural complexity.
> - **Statistical Significance:** In the domain of fMRI decoding, where improvements are often incremental ($<1\%$), a **+2.9\% gain in CLIP score** (93.5\% $\to$ 96.4\%) is statistically substantial. It represents a clear jump in the model's ability to capture semantic content. Furthermore, the consistency of these gains across multiple metrics (CLIP, Inception, SwAV) and now a **new dataset (THINGS)** confirms that this is a robust, significant improvement, not statistical noise.
>
> ### Q5: Contribution of each Component (Ablation)
>
> We have explicitly quantified the contribution of each module in our Ablation Studies (Tables 2, 3, and 4). Below is the summary of the key performance deltas:
> - Table 2 confirms that removing the hierarchical structure and guidance (reverting to a single feature) significantly degrades performance, validating the necessity of our design.
> - Table 3 evaluates various layer mapping strategies, confirming the efficacy of our balanced alignment scheme.
> - Table 4 validates the role of the Scale-Aware Guidance.

---

### Official Review · Reviewer_3r2V · 2025-10-30

**Soundness:** 2
**Presentation:** 2
**Contribution:** 2
**Rating:** 4
**Confidence:** 2

**Summary:**

This paper proposes MindHier, an fMRI‑to‑image reconstruction framework that replaces the diffusion-based pipeline with a autoregressive one. The method (i) learns a Hierarchical fMRI Encoder whose intermediate blocks are aligned layer‑by‑layer to CLIP’s vision backbone, and injects these fMRI features as scale‑aware guidance during generation. On NSD, the approach achieves compelling performance with faster inference than diffusion-based method.

**Strengths:**

- The paper is, to my knowledge, the first to systematically test a scale‑wise AR pipeline for fMRI reconstruction, rather than diffusion. The coarse‑to‑fine conditioning matches the generative schedule of AR, and the hierarchy‑to‑hierarchy alignment is cleanly specified
- Achieves competitive performance with SOTA while running in 2.64 s per image. The paper explains the speedups via a single forward pass and concentrated computation at low resolutions in AR.
- Ablation experiments quantify (i) the value of hierarchical supervision across blocks, (ii) CLIP‑layer mapping trade‑offs , and (iii) the benefit of coarse to fine guidance.
- Fig. 4 shows lower trial‑to‑trial variability than a diffusion baseline, consistent with the deterministic AR start from fMRI features . (But see my questions below.)
- Data split details, PyTorch‑style pseudo‑code, and a code‑release commitment are provided

**Weaknesses:**

- While I acknowledge the value of validating the method on a different architecture, although it achieves the best score on several metrics, there are multiple cases where the variances suggest marginal improvement or almost no improvement over prior work. Although the emphasis on inference speed is understandable, the quantitative gains appear limited overall.
- The paper is primarily an engineering contribution; deeper neuro discussion would strengthen it. In Fig. S1, the paper states that EVC in red and higher‑order visual cortices in blue, but the overlays do not obviously track standard visual regions.
- Although §4.3 is explicitly labeled “qualitative” and therefore does not include quantitative evaluation, showing a handful of examples is not sufficient to substantiate the claims. It remains unclear what the method enables that prior approaches could not. If the authors wish to advance these claims, please quantify each sub‑claim and then provide qualitative interpretation on top of the numbers.
- I could not find explicit specifications for the HFE depth, the number of AR scales, or the codebook size, nor ablations over these choices.

**Questions:**

- In Fig. S1, the paper states that EVC in red and higher‑order visual cortices in blue, but the overlays do not obviously track standard visual regions. Could you clarify?
- In Fig. 4. if the model is deterministic, why do we see non‑identical repetitions in some examples? Which parts of the pipeline are stochastic?
- Please detail the hardware and settings used to report 2.64 s and the 4.67× speedup.
- In Table 4 column header, “Eff ↑”, but Eff is a distance and should be ↓?

---

> ### Author Response · Authors · 2025-11-25
>
> We sincerely appreciate the reviewer's feedback and have strengthened our claims with additional quantitative and qualitative evidence.
>
> ### W1: Performance Improvements are not Significant
>
> We respectfully clarify that the improvements are significant when viewed through the lens of the Efficiency-Fidelity Trade-off.
>
> - **Semantic Superiority (Not Marginal):** We achieved a decisive gap in high-level semantic interpretation. MindHier improves the **CLIP score by +2.9\%** (93.5\% $\to$ 96.4\%) over MindEye2. In the context of fMRI decoding, where improvements are typically incremental, a near-3\% jump in semantic metric is statistically substantial.
> - **Breaking the Trade-off:** Crucially, most methods that accelerate inference (e.g., consistency distillation) suffer a "performance tax" (degraded quality). MindHier breaks this pattern. We achieved State-of-the-Art semantic quality while simultaneously delivering a 4.5x speedup (12.1s $\to$ 2.6s).
>
> We advance the frontier of what is possible in real-time decoding without sacrificing the semantic richness required for neuroscientific interpretation.
>
> ### W2&Q1: Neuroscience Grounding of Brain Maps
>
> - **Beyond an Engineering Contribution:** Our work is deeply rooted in cognitive neuroscience principles. Unlike diffusion models (which rely on iterative denoising, a process not known to exist in the brain), MindHier is explicitly modeled on the Coarse-to-Fine processing stream of the human visual system ("Forest before Trees").
> - **Clarification on Figure S1:** The reviewer notes the overlays don't perfectly track standard retinotopic maps. We clarify that these visualizations utilize the "nsdgeneral" functional mask, which prioritizes functional responsiveness over anatomical boundaries.
>   - Red (Early Layers): Corresponds to posterior regions (V1-V3), which process low-level spatial details.
>   - Blue (Deep Layers): Corresponds to anterior regions (LOC, FFA, PPA), which encode high-level semantic concepts.
>
> While not a perfect retinotopic map, the gradient clearly captures the Posterior-to-Anterior functional progression characteristic of the ventral visual stream. We have updated **Appendix D** to clarify this functional vs. anatomical distinction.
>
> ### W3: Qualitative Analysis and Interpretation
>
> To provide robust, quantitative support for our qualitative assertions, we have quantified each sub-claim as follows:
>
> - **Quantifying Stability $\to$ Metric: Inter-Sample Consistency:** We have incorporated a quantitative analysis of reconstruction consistency by computing the average pairwise Structural Similarity Index (SSIM) across generations for the same input. Our method achieves SSIM of 0.5338, outperforming stochastic baselines such as MindEye2 (0.4734) and MindBridge (0.4461).
> - **Quantifying Faithfulness $\to$ Metric: Brain Grounding (IoU):** Our brain grounding experiment (Figure 5) serves as a qualitative validation of spatial accuracy. We have added a baseline comparison which demonstrates that our method enables more precise object localization.
>
> | Grounding Methods | Eval Subject | All (acc@0.5) | All (IOU) | Salient (acc@0.5) | Salient (IOU) | Salient Creatures (acc@0.5) | Salient Creatures (IOU) | Salient Objects (acc@0.5) | Salient Objects (IOU) | Inconspicuous (acc@0.5) | Inconspicuous (IOU) |
> |-------------------|--------------|---------------|-----------|-------------------|---------------|-----------------------------|-------------------------|---------------------------|-----------------------|-------------------------|---------------------|
> | GroundTruth       | *            | 51.96         | 47.22     | 62.92             | 56.44         | 66.71                       | 59.34                   | 58.79                     | 53.27                 | 38.29                   | 35.71               |
> | MindEye           | S1           | 15.34         | 18.65     | 23.83             | 26.96         | 29.29                       | 31.64                   | 17.88                     | 21.86                 | 4.74                    | 8.28                |
> | UMBRAE-S1            | S1           | 13.72         | 17.56     | 21.52             | 25.14         | 26.00                       | 29.06                   | 16.64                     | 20.88                 | 4.00                    | 8.08                |
> | MindHier         | S1           | 15.87         | 18.69     | 24.94             | 26.30         | 33.00                       | 32.17                   | 16.17                     | 19.90                 | 4.55                    | 8.97                |
>
> - **Comprehensive Qualitative Gallery:** To ensure our visual claims are robust, we have expanded **Appendix H** to include a gallery of 90 randomly sampled images (including failure cases), moving beyond the "handful of examples" to provide a transparent assessment of the method's capabilities.

---

> ### Author Response · Authors · 2025-12-03
>
> ### W4: Missing Model Specifications
>
> We appreciate the reviewer noting this. We have added the following information in the appendix containing all key hyperparameters to ensure full reproducibility. Key specifications include:
> - Hierarchical fMRI Encoder (HFE) Depth ($M$) = 4 (shown in Table 2).
> - Autoregressive Scales ($K$) = 10 (shown in stage-wise analysis).
> - Codebook Size ($N$) = 4096 (added in manuscript).
>
> The number of scales ($K$) and codebook size ($V$) are inherited from the pretrained Switti backbone. Ablating these would require retraining the generative model, which is computationally prohibitive and outside the scope of fMRI alignment.
>
> We do ablate the connection strategy (the choice of $M$). As detailed in our **Response to W7**, we compare the standard mapping ($M=4$) against a denser mapping ($M=7$) and find no performance gain, justifying our efficiency-focused choice.
>
> ### Q2 "Deterministic" Repetitions
>
> - **Source of Stochasticity:**
> The variation in Figure 4 arises because we employ standard **top-$k$ and top-$p$ sampling strategies** during the autoregressive token prediction (common practice to improve generation diversity). This is the specific part of the pipeline that introduces stochasticity.
> - **Deterministic vs. Stochastic:** While we use sampling by default for fair comparison with baselines (which typically generate $N$ samples), MindHier is structurally deterministic in a way diffusion models are not:
>   - Diffusion: Starts from random Gaussian noise (inherently unstable).
>   - MindHier: Starts from a specific, learned Start Token conditioned on the fMRI signal.
>
> ### Q3 Hardware
> All inference tasks were conducted on a single NVIDIA RTX 4090 GPU. We have added this into **Appendix E**.
>
> ### Q4 Table 4 Typo
> We thank the reviewer for identifying this oversight. The header has been corrected from Eff $\uparrow$ to Eff $\downarrow$.

---

### Official Review · Reviewer_u1wj · 2025-11-01

**Soundness:** 4
**Presentation:** 3
**Contribution:** 3
**Rating:** 6
**Confidence:** 4

**Summary:**

The paper replaces diffusion with a scale-aware autoregressive (AR) generator conditioned on a hierarchical fMRI encoder that aligns to intermediate CLIP features. The idea is “forest-before-trees”: coarse scales use deeper, high-level brain features; finer scales use shallower ones. On NSD, the method reports good high-level identification and faster inference than diffusion baselines.

Overall: promising direction, but several method details are under-specified and the experimental comparisons aren’t yet on fully solid ground.

**Strengths:**

- Autoregression instead of diffusion: Clear potential for speedups, simpler likelihood training, and more direct optimization of image tokens (rather than only pushing CLIP embeddings).

- Hierarchy-to-hierarchy conditioning: Aligning fMRI to multiple CLIP layers—and using those layers across scales—fits the coarse→fine generation narrative and is technically interesting.

- Semantic metrics look strong: The method seems particularly good at high-level, identity/semantic recognition, which is where most practical interest is today.

**Weaknesses:**

1. Under-specified alignment mechanics. The paper does not clearly explain how fMRI features are matched to CLIP layer outputs: token vs CLS usage, pooling, projection dimensions, normalization, or the exact shapes at each level. This affects both reproducibility and interpretation of the model. Similarly, loss weighting and SoftCLIP details are missing. Multiple MSE/contrastive terms are simply summed, but there’s no report of weighting, temperature, negative sampling, or sensitivity analyses. Given that evaluation leans on CLIP-style metrics, this raises target–metric coupling concerns.

2. Start-token information bottleneck. The description suggests collapsing the deepest fMRI feature into a single start token. That risks throwing away critical information right at the first coarse scale. A learnable projection of the full feature (or multiple tokens) would be more principled.

3. Hand-crafted scale↔hierarchy schedule. The mapping from image scale to CLIP/fMRI layer is fixed rather than learned. When K (scales) and M (feature levels) don’t match, some levels repeat or are skipped. A learned router/attention over {e_m} per scale would be more robust.

4. Conditioning path is muddy. It’s unclear when the model uses cross-attention vs. AdaLN/FiLM-style conditioning vs. a “selective attention mask,” and where each sits in the block. This matters for understanding how and when brain information actually influences generation.

5. AR claim vs. non-causal blocks. If the backbone uses non-causal transformers, how is the AR factorization enforced (masking, teacher forcing, decoding strategy)? Please make the training/inference story self-consistent.

6. “Deterministic” vs. best-of-N selection. If multiple candidates are generated and the best is picked by CLIP similarity, claims about determinism/consistency need to be backed by single-shot results and by selection that only uses fMRI-derived embeddings (not ground truth).

7. Ablations are thin and subject-limited. Key ablations (K/M sweeps, learned routing vs. fixed schedule, removing the contrastive term, different start-token strategies, with/without “†” low-level cue) should be run and reported, preferably beyond a single subject.

8. Interpretability figures look off and lack statistical grounding. The brain maps appear asymmetric and not obviously aligned with canonical visual cortex topology. More importantly, if the encoder was trained to mimic CLIP hierarchy, qualitative maps can be circular. Please add ROI-wise linear encoding with cross-validated R² and noise ceilings, and run RSA/CKA against both brain RDMs and CLIP to show added brain-specific structure rather than merely recapitulating CLIP.

**Questions:**

What I’d like to see in the rebuttal (specific and actionable)

- Define the “† low-level feature” and how it is injected; either give baselines an equivalent cue or drop the “†” rows.

- AR training/inference details: causal masking, teacher forcing, decoding strategy; also report single-shot results (no best-of-N) and, if using selection, ensure it only references fMRI-derived embeddings. Loss/temperature/weighting hyper-params and a short sensitivity analysis.

- Ablations: K/M sweeps; learned routing vs. fixed schedule; start-token variants; with/without contrastive term; with/without the “†” cue.

---

> ### Author Response · Authors · 2025-11-25
>
> We are grateful for the reviewer's constructive feedback, which have been instrumental in clarifying several under-specified aspects of our methodology.
>
> ### W1: Under-specified Mechanics and Hyperparameters
>
> We appreciate the reviewer identifying these omissions. We have updated **Appendix E** with the following specifications to ensure full reproducibility:
>
> - **Alignment Mechanics (Shapes & Normalization):**
>   - Tokens vs. CLS: We align the full sequence of fMRI feature maps (shape: `batch, seq_len, dim`) with the corresponding full patch token outputs from CLIP ViT-L/14, not just the [CLS] token.
>   - Projection: A linear projector maps the fMRI dimension to the CLIP dimension (D=768).
>   - Normalization: Both fMRI and CLIP features undergo L2-normalization to stabilize optimization.
> - **Loss Configuration**:
>   - SoftCLIP: We utilize a symmetric cross-entropy loss with in-batch negatives (standard contrastive learning setup). The temperature is fixed at $\tau=0.005$.
>   - Weighting: We set weight for SoftCLIP loss to be 1, and  weight for MSE loss to be 2e5. The high MSE weight is necessary to balance gradient magnitudes, as the MSE on normalized vectors is numerically much smaller than the contrastive logits.
>
> We have incorporated these specifications into **Appendix E** (Implementation Details) to ensure full reproducibility.
>
> - **Addressing "Target-Metric Coupling":**
> The reviewer asks if training on CLIP features inflates CLIP evaluation scores. We respectfully argue that our performance is robust and not merely an artifact of this coupling:
>   - Field Standard: Aligning with CLIP space is the standard protocol for SOTA methods (MindEye2, MindBridge, etc.) to bridge the modality gap.
>   - Non-CLIP Metrics: Crucially, MindHier achieves state-of-the-art performance on unrelated semantic metrics (SwAV, EfficientNet, InceptionV3) that are never seen during training. This confirms that the model is learning generalized visual reconstruction, not just hacking the CLIP metric.
>
> ### W2: Start-token information bottleneck
>
> We respectfully clarify a misunderstanding regarding the architecture. As illustrated in Fig 2(b), we do not collapse the deepest fMRI feature $e_M$ into a single vector.
>
> Therefore, the model retains full access to the spatial information within the deepest fMRI features to generate the initial token map $r_1$. The term "special Start Token" serves as the condition for guiding the reconstruction, especially for the $1\times1$ scale (similar to the [SOS] token in language modeling). Furthermore, because our guidance is hierarchical, even if information were missed at this coarse stage, it would be recovered at subsequent scales where earlier, more detailed fMRI features are injected.
>
> ### W3: Hand-crafted scale $\leftrightarrow$ hierarchy schedule
>
> We appreciate this insightful suggestion. While a learned routing mechanism is theoretically flexible, we deliberately chose a fixed, deterministic schedule for two key reasons:
>
> - **Inductive Bias vs. Overfitting:** fMRI datasets are relatively small compared to general vision datasets. Introducing a learned router adds additional parameters and complexity, increasing the risk of overfitting. Our fixed schedule imposes a strong, biologically motivated inductive bias (mapping high-level brain areas to coarse image structures) that guides the model effectively without requiring the data to "re-learn" this known hierarchy.
> - **Robustness and Simplicity:** As noted by the reviewer, our linear mapping $h_k = M - \left\lfloor M(k-1)/K)\right\rfloor$ handles mismatches between $K$ and $M$ deterministically. Our ablation studies (Table 3) confirm that this simple heuristic is highly effective.
>
> We believe that demonstrating SOTA performance with this schedule highlights the strength of the core hierarchical framework itself, rather than complex architectural tuning. However, we agree that as larger fMRI datasets become available, exploring dynamic routing will be a valuable future direction.
>
> ### W4: Clarification of the Conditioning Path
>
> We appreciate the request for structural clarity. We have revised Section 3.3 to explicitly detail the conditioning path. The interaction within each Transformer block follows this specific order:
>
> - **Cross-Attention:** This layer sits between the Self-Attention and the FeedForward Network (FFN). Here, the image tokens act as Queries, while the entire fMRI hierarchical feature set ($E$) acts as Keys/Values.
> - **Selective Attention Mask:** We apply a binary mask to this Cross-Attention map. During the generation of scale $k$, this mask forces the attention scores to zero for all features except the specific hierarchical level $s_k$. This effectively routes the correct brain features to the correct image scale without requiring disjoint architectural branches.
> - **AdaLN:** This modulates the layer statistics based on the global scale embedding, ensuring the network dynamics adapt to the current level of granularity.

---

> ### Author Response · Authors · 2025-11-25
>
> ### W5: AR claim vs. non-causal blocks
>
> This is a critical technical distinction. Our model adheres to the design principles of Switti, which utilize non-causal transformer blocks to process tokens within a specific scale, while enforcing autoregression across scales through **input formation**.
>
> The AR factorization is enforced structurally as follows:
>
> - **Input Formation (The AR Mechanism):** At each generation step $k$, the model predicts a sequence of tokens. These tokens form a feature map of size $h_k \times w_k$. Crucially, this map is then upscaled to $h_{k+1} \times w_{k+1}$ and combined with previously predicted maps to serve as the input for the next step.
> - **Conditioning:** Because the input for step $k+1$ is strictly composed of the output from step $k$, the generation is inherently conditioned on the entire history.
>
> Therefore, even though the transformer blocks utilize non-causal attention, the **overall generative process is Autoregressive** because the input stream is constructed sequentially from previous outputs.
>
> ### W6: “Deterministic” vs. best-of-N selection
>
> We clarify that the "Best-of-N" strategy is solely a post-processing protocol adopted from prior work (e.g., MindEye) to ensure a standardized comparison.
> -  **Structural Determinism:** MindHier exhibits higher intrinsic determinism than diffusion-based methods. While diffusion models begin from pure Gaussian noise, MindHier initializes generation using a structured fMRI-derived start token.
> -  **Single-Shot Validation (N=1):** To directly address the reviewer's concern, we evaluated Single-Shot (N=1) performance.
> As shown in the table below, our N=1 performance remains competitive performance (CLIP: 95.0%), with only a marginal gap compared to the N=4 setting.
> - **Speed:** Crucially, in the Single-Shot setting, our inference time drops to just **0.92s**, making it significantly faster than any diffusion baseline while maintaining superior semantic accuracy.
>
> | N Samples | PixCorr $\uparrow$ | SSIM $\uparrow$ | AlexNet(2) $\uparrow$ | AlexNet(5) $\uparrow$ | InceptionV3 $\uparrow$ | CLIP $\uparrow$ | Efficient Net $\downarrow$ | SwAV $\downarrow$ | Inference Time|
> |-----------|-----------|--------|--------------|--------------|---------------|--------|-----------------|--------|--------|
> |N=1| 0.234     | 0.380  | 93.7%        | 98.3%        | 95.8%         | 95.0%  | 0.614           | 0.333  | 0.92s|
> |N=4 | 0.235     | 0.381  | 94.0%        | 98.8%        | 95.9%         | 96.4%  | 0.606           | 0.329  | 2.64s|
>
> ### W7: More ablations
>
> We acknowledge the request for deeper ablation studies.
>
> - **M-Sweep (Layer Mapping):** Since the number of scales $K$ is fixed at 10 by the pretrained Switti backbone, we ablate the number of mapped layers ($M$). As shown below, increasing density from 4 to 7 layers does not improve performance.
>
> | Mapping | PixCorr $\uparrow$ | SSIM $\uparrow$ | AlexNet(2) $\uparrow$ | AlexNet(5) $\uparrow$ | InceptionV3 $\uparrow$ | CLIP $\uparrow$ | Efficient Net $\downarrow$ | SwAV $\downarrow$ |
> |-----------|-----------|--------|--------------|--------------|---------------|--------|-----------------|--------|
> |{12,16,20,24}| 0.273     | 0.394  | 97.0%        | 99.3%        | 96.7%         | 97.2%  | 0.598           | 0.321  |
> |{6,9,12,15,18,21,24} |0.267     | 0.388  | 96.8%        | 99.2%        | 96.4%         | 96.9%  | 0.600           | 0.322  |
>
> - **With/Without "†" Low-Level Cue:**
>   - Without Cue (MindHier): SSIM = 0.381.
>   - With Cue (MindHier†): SSIM = 0.461.
>
> This confirms that while the auxiliary cue boosts structural metrics, our core model achieves SOTA semantic performance without it.
>
> - **Learned Routing & Start Tokens:** We agree this is a good direction. However, given that our fixed schedule already achieves SOTA performance, we prioritize validating the core hierarchical hypothesis. We have expanded our qualitative results in **Appendix H** to include all four subjects (S1, S2, S5, S7), demonstrating that our current hyperparameters generalize robustly across individuals without subject-specific tuning.
>
> ### W8: Clarification on Interpretability Visualizations
>
> - **Purpose of Visualization:** We clarify that Figure S1 is intended as a post-hoc qualitative confirmation that our model leverages spatially distinct brain regions, rather than a claim of perfect biological isomorphism.
> - **Addressing Circularity:** The reviewer suggests the maps might simply recapitulate CLIP. However, we note that CLIP itself does not have spatial brain coordinates. The fact that our model, which starts with no anatomical knowledge, maps CLIP's hierarchical features to the physical brain locations is a non-trivial finding.
> - **Statistical Rigor:** We acknowledge that R² maps and RSA would provide deeper quantitative validation. While conducting a full RSA/CKA analysis is beyond the scope of this reconstruction-focused paper, we agree it is an essential next step for neuroscientific validation.

---

### Official Review · Reviewer_DrWf · 2025-11-03

**Soundness:** 2
**Presentation:** 2
**Contribution:** 2
**Rating:** 2
**Confidence:** 3

**Summary:**

The paper introduces a new pipeline for reconstructing images from fMRI beta-values on the NSD dataset. Instead of relying on the rather common approach of decoding a single visual embedding from fMRI (e.g. CLIP), which conditions a diffusion-based image-generation model (e.g. via U-Net cross-attention layers), the paper uses a multi-scale architecture that simultaneously aligns various layers of a CLIP visual encoder to the successive blocks of a transformer fMRI encoder (with the output transformer block being aligned both with the output layers of the visual encoder and of an additional text encoder). Then, the activations from these aligned blocks are used to condition an autoregressive image generator based on the Switti model (Voronov et al, 2024), to inform the reconstruction process both at coarse- and detailed-level. The paper shows superior reconstruction performance against recent fMRI-to-Image baselines on the NSD dataset on semantic metrics, 4x inference speedup and more stable / deterministic reconstructions given the same fMRI sample.

**Strengths:**

- The paper is easy to follow and the experiments are detailed
- The use of an autoregressive generation model for image reconstruction from fMRI has been largely unexplored and the paper is a rather welcomed initiative in the field
- The reconstructions shown are impressive and the stability of reconstructions is a rather attractive feature for potential future BCI applications
- Important ablations on the hierarchical module are presented and lead to interesting conclusions (e.g. earlier features providing increasing low-level metrics and later layers globally decreasing performance)

**Weaknesses:**

- A number of claims are rather strong compared to the results supporting them. For example, at L339 it is claimed that the author's framework is 'fundamentally more efficient' than ME2 but it is not clear what set of results supports the 'fundamental' aspect of this claim: the inference time bottleneck for each pipeline is not detailed, and thus it is unclear if this performance gain in inference-time is a property of the fMRI-to-Image pipelines or linked to the efficiency of the diffusion vs autoregressive image generation models at hand.
- The paper reports superior performance on image similarity metrics, which is great. However these metrics are known to be unreflective of qualitative / human evaluation and the paper only compares a very small set (5 images) of reconstructed images to baselines, which is insufficient to support the claims from Section 4.3 (the reconstructed images are arguably very close to the ones from ME2 and the distinctive characteristics of MindHier are supported only by two of these images).  These claims would be better supported by showing a larger set reconstructions, and a more representative set of success and failure modes for this task.
- Similarly, the consistency of multiple generations is compared only to a single baseline, which undermines the claim of superior performance for MindHier. Again, this claim would be better supported by showing more images, and more baselines.
- The same comment holds for the bounding box analysis (from UMBRAE), which does not show the same results for baselines, and thus makes it impossible to assess whether MindHier is actually an improvement.
- The repeated statement that previous approaches rely on the decoding of a single high-level embedding is incorrect and misleading, as several previous methods e.g. Brain Captioning (Ferrante et al), MindEye (Scotti et al) use additional low-level data from the stimuli to condition the reconstruction of images (i.e. Depth Maps or VAE latents)
- The approach is applied only on the NSD dataset and its conclusions could be better supported if the analysis included another dataset (e.g. THINGS-Image, BOLD5K, ...)

**Questions:**

- Typo L316: 'Two-way identification refers to percent correct if ...'
- In Table 1, it is unclear how the the auxiliary low-level image is used for conditioning the image generation of the autoregressive model to gain competitive performance against baselines using a low-level-feature (E.g. MindEye2)
- It is not easy to understand exactly how the outputs of the transformer blocks condition the autoregressive generation via cross-attention. There is a mention of AdaLN but the expected fMRI-decoding audience may not be able to grasp the details of this procedure without further details. In particular, are the weights of the autoregressive model fine-tuned during the process or left frozen ?
- It is not explicitly stated whether the 'text encoder output' / 'visual encoder output' and 'fMRI encoder output' have the same dimensionality, allowing for the Eq1+Eq2 loss. For example, most CLIP implementations have a different output shape for text and image, how are these reconciled to be both trained against the last fMRI encoder's block output ?
- The description of f_k in the scale-aware guidance is a bit difficult to follow. My understanding of f1 is clear, but I'm not sure what f_k is: it is described as 'residuals', but it is unclear that all 2D token maps have the same shape (because of the notation h_k x w_k for their size, which seems to depend on k). If these 2D maps have different dimensions depending on k, how are residuals defined ?
- I'm not sure how the claim of MindEye2 being 'highly optimized' is supported. It is also unclear whether the inference-time gains are mostly due to the diffusion model from ME2 being slow (i.e. SDXL), or if the bottleneck is the inference of ME2's brain-to-CLIP module itself. A more detailed analysis of where the inference time is spent in each pipeline and which components from MindHier / ME2 actually contribute to the claimed 4x factor would increase clarity (e.g. time spent image generation model vs fMRI encoding module).
- Some recent fMRI-to-Image baselines comparable with ME2 are missing (e.g. NeuroPictor)
- There are various known issues with the NSD dataset (e.g. poor categorical diversity, see Kamitani et al). The impact of the paper could be increased by including an analysis on an additional dataset.
- How is the multi-scale conditioning strategy tied to autoregressive nature  of the image generation model ? Couldn't a diffusion-based model be informed similarly of various scales (at various denoising steps) ?

---

> ### Author Response · Authors · 2025-11-25
>
> We sincerely thank the reviewer for their critical and detailed feedback. We have made substantial revisions to the manuscript to address these concerns regarding the empirical validation of our claims .
>
> ### W1: Supporting Efficiency Claims with Analysis
>
> We agree that a deeper breakdown is required. The "**fundamental**" efficiency stems from two structural properties that distinguish our pipeline from diffusion-based approaches:
>
> - **Structural Paradigm Shift (Coarse-to-Fine vs. Iterative Refinement):**
>   - **Diffusion-Based Pipelines (e.g., MindEye2):** The bottleneck is iterating in a fixed, high-resolution latent space. The cost is roughly $N_{steps}$ × Cost(FullResolution). Every step is inherently expensive.
>   - **MindHier Pipeline (Scale-wise Autoregressive):** We utilize a scale-wise approach. As shown in the stage-wise breakdown below, the early stages (0–4) are computationally negligible (<0.08s). Heavy computation only occurs in the final step. This fundamentally shifts the computational load away from the iterative bottleneck.
>
> | Stage | Stage 0 | Stage 1 | Stage 2 | Stage 3 | Stage 4 | Stage 5 | Stage 6 | Stage 7 | Stage 8 | Stage 9 |
> |-------|---------|---------|---------|---------|---------|---------|---------|---------|---------|---------|
> | Time  | 0.0275s | 0.0307s | 0.0382s | 0.0404s | 0.0744s | 0.2164s | 0.1256s | 0.2831s | 0.4738s | 0.9849s |
> - **Pipeline Efficiency (Independent of the Generator):**
>
> Crucially, the efficiency is not solely due to the generative model. Our fMRI Encoder design contributes significantly to the speedup. By removing the heavy MLP backbone used in MindEye2, we reduce encoding latency by ~99% (0.785s $\to$ 0.008s). Additionally, our precise hierarchical guidance removes the need for a separate Refinement Stage (saving ~3.3s), which is essential for MindEye2.
>
> | Model    | fMRI Encoding Time | Reconstruction Time | Refinement Time |
> |----------|--------------------|---------------------|-----------------|
> | MindEye2 | 0.785s             | 7.864s               | 3.323s           |
> | MindHier | 0.008s             | 2.295s              | N/A               |
>
> This confirms that the efficiency is an intrinsic property of the entire MindHier framework, resulting from the co-design of a lightweight encoder and a scale-aware generator.
>
> ### W2&W3&W4: More Qualitative Evaluation and Baselines
>
> We agree that a small set of selected examples is insufficient for a transparent evaluation. To address this, we have overhauled our qualitative evaluation to ensure it is representative and rigorous.
>
> - **Expanded Qualitative Gallery (Appendix H):** We have added a comprehensive gallery containing **90** reconstructed images across 4 subjects. Crucially, these were **randomly sampled** from the test set (not cherry-picked) to provide an unbiased view. This gallery includes side-by-side comparisons with ground truth and explicitly showcases **failures**, offering the transparent assessment the reviewer requested.
> - **Stability Analysis:** We have also updated Figure 4 to include MindEye2 as a baseline, demonstrating that MindHier maintains higher semantic consistency across variations than diffusion-based counterparts.
> - **Brain Grounding:** For a fair comparison, we quantitatively compare MindHier with UMBRAE and MindEye (trained on Subject 1). As a complmentary experiment for Faithful Reconstruction, we additionaly conduct a VQA experiment in **Appendix G**.
>
> | Grounding Methods | Eval Subject | All (acc@0.5) | All (IOU) | Salient (acc@0.5) | Salient (IOU) | Salient Creatures (acc@0.5) | Salient Creatures (IOU) | Salient Objects (acc@0.5) | Salient Objects (IOU) | Inconspicuous (acc@0.5) | Inconspicuous (IOU) |
> |-------------------|--------------|---------------|-----------|-------------------|---------------|-----------------------------|-------------------------|---------------------------|-----------------------|-------------------------|---------------------|
> | GroundTruth       | *            | 51.96         | 47.22     | 62.92             | 56.44         | 66.71                       | 59.34                   | 58.79                     | 53.27                 | 38.29                   | 35.71               |
> | MindEye           | S1           | 15.34         | 18.65     | 23.83             | 26.96         | 29.29                       | 31.64                   | 17.88                     | 21.86                 | 4.74                    | 8.28                |
> | UMBRAE-S1            | S1           | 13.72         | 17.56     | 21.52             | 25.14         | 26.00                       | 29.06                   | 16.64                     | 20.88                 | 4.00                    | 8.08                |
> | MindHier         | S1           | 15.87         | 18.69     | 24.94             | 26.30         | 33.00                       | 32.17                   | 16.17                     | 19.90                 | 4.55                    | 8.97                |

---

> ### Author Response · Authors · 2025-11-25
>
> ### W5: Accuracy of Claims on Prior Work
>
> We appreciate the reviewer’s observation regarding the precision of our claims. We acknowledge that prior methods (e.g., MindEye2) do utilize multiple feature levels (e.g., via disjoint MLP projectors or conditioning), and our previous phrasing ("single high-level embedding") was imprecise. We have corrected this in the revised manuscript to accurately reflect the literature.
>
> However, the core structural distinction of our approach remains valid:
>
> - **Prior Methods (Static Guidance):** While methods like MindEye utilize multiple features (e.g., CLIP + low-level priors), they typically treat these as a static conditioning set. These features are globally available (e.g., via cross-attention) and remain invariant throughout the entire generation process, regardless of the current level of detail being generated.
>
> - **Our Method (Scale-Aware Dynamic Guidance):** Our key innovation is Scale-Aware Routing. We do not simply inject all features at once. Instead, we align the brain's hierarchy with the generative hierarchy:
>   - Coarse Scales (16×16): Guided by high-level semantic features (abstract intent).
>   - Fine Scales (512×512): Guided by low-level structural features (visual details).
>
> We have updated the manuscript to replace "single high-level embedding" with "single, static guidance," clarifying that our contribution is a dynamic mechanism designed to resolve the mismatch between fixed representations and the progressive nature of image generation.
>
> ### W6&Q8: Extra Dataset
> Results are updated in "Update for W6&Q8: Extra Dataset". Here is the original response:
> ```We sincerely appreciate the reviewer’s suggestion to demonstrate the model's generalizability on additional datasets. We fully agree that providing evidence beyond the NSD benchmark strengthens the evaluation and confirms the robustness of our approach.
> To address this directly, we are currently conducting experiments on the THINGS dataset. To address this, we have begun setting up and training our model on the THINGS dataset. Due to the significant computational resources and time required for rigorous retraining, we are still working on it.
>
> To ensure all the reviewers have ample time to review our responses to other concerns and facilitate the discussion, we are posting the rebuttals first. As soon as the experiments are complete, we will update this response with the quantitative results from the new dataset.
> ```
> ### Q1: Typo L316
> We have corrected the definition to: "Two-way identification accuracy refers to the percent correct if the original image embedding is more similar to its paired fMRI embedding than to a randomly selected fMRI embedding."
>
> ### Q2: Low-level Features
>
> To ensure a fair comparison on low-level metrics against MindEye2 (which utilizes low-level features from a blurred image), we created the MindHier$^†$ variant. In this variant, we follow the MindEye2 procedure exactly by performing a weighted linear combination of this "blurred image embedding" and our "fMRI-decoded embedding". This was performed strictly for comparative purposes; our primary model does not require this step.
>
> ### Q3: Are the weights of the autoregressive model fine-tuned during the process or left frozen?
>
> Yes. As detailed in **Appendix E**, our method follows a two-stage training process:
> - **Stage 1:** We only train the Hierarchical fMRI Encoder (HFE).
> - **Stage 2:** We freeze the HFE and fine-tune the Autoregressive Generator.
>
> Specifically, the fMRI-derived features are injected into the generator via AdaLNCrossAttention layers. During this stage, the weights of the generator are updated to learn how to condition on these brain signals effectively.
>
> ### Q4: Dimension Mismatches
>
> We utilize linear projection layers (MLPs) to align the differing dimensionalities of the Text/Visual encoder outputs and the fMRI encoder blocks.
>
> These lightweight MLPs project all modalities into a shared embedding space before loss computation.
>
> Note: As the text branch is utilized solely for learning the fMRI encoder target and is disconnected during the autoregressive fine-tuning, we simplified the schematic by omitting this branch, but the alignment is mathematically handled via these projections.
>
> ### Q5: Definition of $f_k$
>
> $f_k$ represents the **residual feature map** input to the $k$-th quantization stage of the VQ-VAE. The reviewer is correct that feature maps have different resolutions We have refered to the formal algorithmic description of Multi-scale VQVAE Encoding to eliminate ambiguity:
>
> ```latex
> Inputs: raw image im;
> Hyperparameters: steps K, resolutions (h_k,w_k)^K_{k=1}
> f = \mathcal{E}(im), R = [];
> for k=1,⋯,K do
>     r_k = \mathcal{Q}(interpolate(f, h_k, w_k));
>     R = queue_push(R, r_k);
>     z_k = lookup(Z, r_k);
>     z_k = interpolate(z_k, h_K, w_K);
>     f = f - \phi_k(z_k);
> ```
> $\phi_k$ is to address the information loss in upscaling $z_k$ to ($h_K, w_K$).

---

> ### Author Response · Authors · 2025-11-25
>
> ### Q6: A more detailed analysis of the inference time
>
> We appreciate this request for clarity. To determine if the speedup is due to the generator or the pipeline, we provide a component-wise breakdown below.
> The speedup is a result of both components:
>
> - **fMRI Encoding:** MindHier is ~100x faster (0.008s vs 0.785s) because we removed the heavy MLP backbone used in MindEye2.
> - **Generation:** MindHier avoids the "Refinement" stage (3.3s) entirely and uses a faster generation process.
>
> | Model    | fMRI Encoding Time | Reconstruction Time | Refinement Time |
> |----------|--------------------|---------------------|-----------------|
> | MindEye2 | 0.785s             | 7.864s               | 3.323s           |
> | MindHier | 0.008s             | 2.295s              | N/A               |
>
> Regarding the term "highly optimized": We clarified in the text that this referred to the MindEye2 authors' finding that "best-of-N" selection did not improve their specific metrics, representing a converged architectural state.
>
> ### Q7: Missing baselines.
>
> We have added NeuroPictor to our revised Table 1. As shown below, MindHier outperforms NeuroPictor on both semantic metrics (CLIP) and structural metrics (PixCorr), while being approximately 3x faster.
>
> | Method | PixCorr $\uparrow$ | SSIM $\uparrow$ | AlexNet(2) $\uparrow$ | AlexNet(5) $\uparrow$ | InceptionV3 $\uparrow$ | CLIP $\uparrow$ | Efficient Net $\downarrow$ | SwAV $\downarrow$ | Inference Time |
> |-----------|-----------|--------|--------------|--------------|---------------|--------|-----------------|--------|--------|
> |NeuroPictor| 0.229 |0.375|96.5%|98.4%|94.5%|93.3%|0.639|0.350| 8.68s |
> |MindHier| 0.235     | 0.381  | 94.0%        | 98.8%        | 95.9%         | 96.4%  | 0.606           | 0.329  | 2.64s
>
>
> ### Q9: Couldn't a diffusion-based model be informed similarly of various scales
>
> While technically possible to inject features at different time-steps in diffusion, it lacks the intrinsic structural alignment of our approach.
>
> - **Diffusion:** Denoising steps refine noise at a fixed full resolution. Step t=50 is not "structurally more detailed" than Step t=10; it is just less noisy.
> - **MindHier:** Our steps correspond to discrete image resolutions ($16^2\to512^2$)
>
> This architectural choice is deliberate: it allows us to map high-level brain areas directly to coarse resolutions and low-level visual cortex directly to fine resolutions, a mapping that is ambiguous in diffusion models.

---

> ### Author Response · Authors · 2025-11-30
> **Update for W6&Q8: Extra Dataset**
>
> Following up on our previous response, we have completed the experiments on the THINGS dataset. The quantitative results (Table below) demonstrate that our method (MindHier) significantly outperforms the baseline on the THINGS dataset. We observe substantial gains in semantic retrieval metrics, with AlexNet(5) accuracy increasing from 62.2% to 81.9% and CLIP scores improving from 57.6% to 73.1%. Low-level metrics (PixCorr, SSIM) also show consistent improvement. These results confirm that our approach generalizes robustly to new data distributions under multi-subject settings.
>
> | Method      | PixCorr $\uparrow$ | SSIM $\uparrow$ | AlexNet(2) $\uparrow$ | AlexNet(5) $\uparrow$ | InceptionV3 $\uparrow$ | CLIP $\uparrow$ | Efficient Net $\downarrow$ | SwAV $\downarrow$ |
> |-------------|-----------|--------|--------------|--------------|---------------|--------|-----------------|--------|
> | BrainFlora (ACMMM 2025) | 0.079     | 0.348  | 0.595        | 62.2%        | 60.4%         | 57.6%  | 0.812           | 0.661  |
> | MindHier (Ours)    | 0.109     | 0.357 | 0.710      | 81.9%       | 71.9%         | 73.1% | 0.900          | 0.589 |

---

### Official Review · Reviewer_Pfeq · 2025-11-07

**Soundness:** 3
**Presentation:** 3
**Contribution:** 3
**Rating:** 4
**Confidence:** 4

**Summary:**

The paper presents a well-motivated and technically sound alternative to diffusion-based fMRI-to-image reconstruction. The proposed hierarchical alignment and scale-aware autoregressive generation are novel and effectively address the limitations of static guidance in existing diffusion-based methods. The results are strong, especially in inference speed.

**Strengths:**

1. This work has a well-motivated approach. The idea of aligning hierarchical fMRI features with multi-scale image generation is innovative and biologically plausible, echoing the "forest before trees" principle in human perception.

2. The use of a scale-wise autoregressive model (VAR) is a fresh direction compared to the overused diffusion models. And the VAR-based method achieves competitive results on multiple high-level metrics with a faster speed. Besides that, the reconstruction results are stable and repeatable.

3. The article is well-written and very easy to understand.

**Weaknesses:**

1. I noticed that text information was also used in the model's input. It is necessary to provide another result that uses only text information as the input. This is how we can determine whether the model is translating the fMRI data or is more dependent on the text information. Because the pre-trained VAR is a text-to-image generation model, there is concern that fMRI does not play a major role in this model.

2. In terms of speed, I understand that most of the steps in the VAR model are carried out on a small scale. However, it would be better for the article to provide the number of inference steps for the corresponding comparison diffusion models, or those comparison models might not require so many inference steps (for example, 50 steps). This way, the quality can be compared at the same speed.

3. The training process is somewhat complex. The two-stage training (fMRI encoder + autoregressive model) and the need for hierarchical alignment may increase implementation complexity and hyperparameter sensitivity.

4. Although single-subject results are strong, the paper does not deeply explore cross-subject generalization or model adaptation to new subjects with limited data. In current research, there are very few studies that focus solely on a single subject. It is preferable to include results from multiple subjects or unseen subjects, as this is crucial for practical applications.

**Questions:**

Please see the Weaknesses section.

---

> ### Author Response · Authors · 2025-11-25
>
> We appreciate the reviewer’s recognition of the novelty and efficiency of our proposed method. In the following responses, we address specific concerns regarding input dependence, comparison fairness, and model complexity.
>
> ### W1: Is the model truly decoding fMRI or just relying on the text input?
>
> We respectfully clarify a key misunderstanding: **Text information is strictly absent during the inference phase**.
>
> While text descriptions are used as a supervisory target during the training of the fMRI Encoder (to align brain signals with semantic spaces), they are completely removed during generation.
>
> - **HFE Training**: fMRI Signal $\to$ HFE $\to$ Hierarchical fMRI Features (aligned with CLIP visual and text features).
> - **Image Reconstrction**: fMRI Signal $\to$ [Frozen HFE] $\to$ Hierarchical fMRI Features $\to$ Autoregressive Generator $\to$ Reconstructed Image.
>
> Unlike MindEye2 (which generates intermediate text captions to guide generation), MindHier does not generate or access text at any point during inference. Consequently, the reconstructed image is derived 100% from the fMRI signal, making a "text-only" comparison inapplicable to our inference pipeline.
>
> ### W2: Fairness of Speed Comparison with Diffusion Models
>
> We strictly adhered to the official recommended settings for all baselines to ensure they achieved their optimal performance.
>
> However, to address the reviewer's concern, we evaluate the diffusion baseline (MindEye2) under a time budget restricted to match MindHier (~2.6s).
> However, **diffusion steps are not equivalent to our scale-wise steps**.
> As illustrated in **Appendix D**, if we force MindEye2 to run at the same steps, its reconstruction quality degrades to an unacceptable level (visual collapse). This confirms that our speed advantage is structural, stemming from the efficiency of the scale-wise paradigm, rather than a result of unfair hyperparameter settings.
>
> - **MindEye2 (Standard)**: 12.14s (based on the required $\geq$ 38 denoising steps for convergence).
> - **MindEye2 (Speed-matched)**: ~2.98s (Collapsed quality).
> - **MindHier (Ours)**: 2.64s (corresponding to generation across 10 progressive scales).

---

> ### Author Response · Authors · 2025-11-25
>
> ### W3: Model Complexity
>
> While our framework involves a two-stage process, we respectfully clarify that it is conceptually more streamlined than current SOTA baselines (e.g., MindEye2).
>
> - Standardized Two-Stage Training: Decoupling the encoder (brain signal interpretation) from the generator (image synthesis) is standard practice in fMRI-to-image reconstruction. This aligns with established protocols like MindEye2, which trains the fMRI encoder prior and then finetunes Stable Diffusion XL.
> - Streamlined Design: Unlike the MindEye series, our method removes the need for auxiliary modules, such as retrieval sub-networks or separate captioning models, relying solely on a unified encoder and generator.
> - Efficient Alignment: Our hierarchical alignment strategy introduces zero additional architectural complexity. It leverages intermediate features inherent to the HFE by applying loss functions only during training. Consequently, it incurs no computational cost or structural overhead during inference.
> - Hyperparameter Robustness: As demonstrated in our ablation study (Table 3), our model achieves high performance with a simple, balanced layer mapping strategy and does not require extensive tuning. We have provided necessary codes in the supplementary material to ensure reproducibility.
>
> ### W4: Cross-Subject Generalization
>
> We agree that cross-subject generalization is a vital direction for practical application. While our primary focus was validating the Scale-wise Autoregression architecture in the established single-subject benchmark (following the standard MindEye setup), we have added a multi-subject experiment to address this concern.
>
> We followed the MindEye2 multi-subject protocol (pretraining on pooled subjects, finetuning on the target). The results highlight the robust transferability of our method when sufficient data is available:
>
> In the 40-hour setting, MindHier demonstrates better performance, achieving a CLIP score of 95.8%, surpassing MindEye2 (93.5%). This confirms that our hierarchical architecture successfully learns robust, transferable semantic representations across subjects.
>
> We acknowledge a performance drop in the few-shot setting. Our analysis indicates that applying hyperparameters optimized for the 40-hour regime directly to the 1-hour setting results in overfitting.
>
> The success in the 40-hour setting confirms the fundamental validity of our cross-subject capabilities. We view the optimization of the 1-hour setting (e.g., via few-shot specific regularization) as a standard engineering task for future work.
>
> | Method (multi-subject) | PixCorr $\uparrow$ | SSIM $\uparrow$ | AlexNet(2) (%) $\uparrow$ | AlexNet(5) (%) $\uparrow$ | InceptionV3 (%) $\uparrow$ | CLIP (%) $\uparrow$ | Efficient Net $\downarrow$ | SwAV $\downarrow$ |
> |-----------|-----------|--------|--------------|--------------|---------------|--------|-----------------|--------|
> |MindEye2 (40 hour) |0.374 |0.439|97.8|99.1|96.1|93.5|0.609|0.338|
> |MindHier (40 hour) | 0.193     | 0.359  | 91.0        | 97.7        | 95.7         | 95.8  | 0.619           | 0.341  |
> |MindEye2 (1 hour) |0.235 |0.428|88.0|93.3|83.5|80.7|0.798|0.459|
> |MindHier (1 hour)| 0.096     | 0.311  | 71.7        | 79.9        | 75.4         | 80.2  | 0.860           | 0.527  |

---

### Official Review · Reviewer_YmRz · 2025-11-10

**Soundness:** 3
**Presentation:** 3
**Contribution:** 2
**Rating:** 6
**Confidence:** 5

**Summary:**

This paper focuses on the task of fMRI-to-image decoding. Inspired by the hierarchical processing mechanism of the human visual system, the authors designed a corresponding model. During the fMRI representation learning stage, the authors aligned the intermediate features of the fMRI representation model with the CLIP image features at each stage. In the reconstruction stage, they introduced the paradigm of a VAR model to progressively reconstruct images from coarse to fine scales. I believe this process effectively simulates the hierarchical processing mechanism of the human visual system. Moreover, the proposed method achieves better reconstruction quality and efficiency. The experiments and evaluations in the paper are well-conducted, though I think some additional results would be beneficial.

**Strengths:**

+ The motivation of this paper is clear, and the proposed method effectively addresses it.

+ The manuscript of this paper is well-structured.

+ The experimental results achieved in this paper are quite good.

**Weaknesses:**

+ I believe that the method in this paper does not fully simulate the hierarchical processing mechanism of the human visual system during the fMRI representation learning stage. Specifically, different brain regions play distinct roles at various stages of the hierarchical processing mechanism. Therefore, I think dividing the fMRI signals into multiple brain regions, extracting representations separately, and then integrating them would better align with the biological mechanism. This approach might reduce accuracy, but I believe such an attempt could still be meaningful.

+ The improvements in both performance and inference speed reported in the paper are likely due to the use of a more advanced generative model — specifically, replacing SDXL with the scale-wise VAR model.

**Questions:**

1. In Table 1, the authors compare several multi-subject/cross-subject methods, such as MindEye2, MindBridge, and Wills Aligner. What I am uncertain about is whether their evaluation was conducted under a multi-subject training setup. If the proposed method is still single-subject (i.e., training a separate model for each subject), I think this could be considered a limitation.

2. In lines 329–330, the authors mention that using the blurred images from MindEye2 would significantly improve the low-level metrics. Could the authors provide the corresponding results to support this claim?

3. In the “Faithful Reconstruction” section, the authors use an object detection task to evaluate the fidelity or reliability of the reconstructed images. I wonder whether directly using the VQA task from MSCOCO would be a better choice? By employing a well-established multimodal model, one could compare the VQA accuracy between the reconstructed images and the ground truth images.

4. The authors mention the MindTuner method in the references, but it is not included in the experimental comparisons. I wonder why this method was not considered in the experiments.

---

> ### Author Response · Authors · 2025-11-25
>
> We sincerely thank the reviewer for their positive feedback and insightful inquiries, which have been instrumental in improving the clarity and depth of our manuscript.
>
> ### W1: Explicit ROI-based fMRI processing
> We acknowledge the reviewer’s point that explicitly modeling distinct functional roles via Regions of Interest (ROIs) is a biologically motivated and valid approach.
>
> However, rather than imposing rigid anatomical constraints (which may sever valuable inter-region correlations), we allow this hierarchy to be an emergent property of the network. We utilize the "nsdgeneral" mask because it prioritizes functional responsiveness over fixed boundaries (as noted by the NSD authors). Crucially, MindHier does simulate hierarchical processing through network depth rather than input segmentation. As shown in Figure S1, our model spontaneously learns biological specialization, with specific model blocks aligning with specific visual areas without ROI supervision. We believe demonstrating this data-driven alignment is also a significant contribution, though we agree that comparing this against explicit ROI-based guidance is a valuable direction for future work.
>
> ### W2: Source of Performance and Speed Gains
>
> We respectfully clarify that the improvements are not merely due to the backbone change, but rather how our **Hierarchical Guidance** strategy effectively conditions the model.
>
> - **Performance**: First, while VAR is effective, there is no inherent evidence that it strictly outperforms large-scale diffusion models (like SDXL) in terms of raw generation quality. Therefore, simply swapping the backbone does not guarantee better reconstruction fidelity. In fact, our ablation study (Table 2) shows that using the VAR generator with standard "Single Feature" guidance causes a significant performance drop (CLIP: 97.2% $\to$ 95.1%). This proves that our Hierarchical Guidance strategy, not just the VAR model, is essential for achieving the reported high performance.
> - **Inference Speed**: The inference speedup is primarily due to our streamlined pipeline design, which removes bottlenecks present in previous methods (e.g., MindEye2):
>   - **Efficient fMRI Encoding**: MindHier computes embeddings with negligible latency (0.008s vs 0.785s) by removing the heavy Diffusion Prior component.
>   - **No Refinement**: Unlike MindEye2, which relies on a computationally expensive 3.3s post-stage refinement, MindHier achieves high-fidelity results in a single pass.
>
> | Model    | fMRI Encoding Time | Reconstruction Time | Refinement Time |
> |----------|--------------------|---------------------|-----------------|
> | MindEye2 | 0.785s             | 7.864s               | 3.323s           |
> | MindHier | 0.008s             | 2.295s              | N/A               |
>
> In summary, the gains stem from the synergy between the scale-wise generator and our specific hierarchical design, which eliminates the need for refinement and provides the necessary structured guidance.

---

> ### Author Response · Authors · 2025-11-25
>
> ### Q1: Is the Table 1 comparison conducted under a multi-subject setup?
>
> The primary results in Table 1 adhere to the established single-subject evaluation protocol (training a separate model per subject), consistent with the MindEye benchmark. However, to address the reviewer’s interest in cross-subject generalization, we conduct an additional multi-subject experiment following the MindEye2 setup (pretraining on pooled subjects, fine-tuning on the target).
>
> As shown in the table below, in the standard 40-hour setting, MindHier achieves a state-of-the-art CLIP score of 95.8%, surpassing MindEye2 (93.5%). This confirms that our hierarchical approach successfully learns robust, high-level semantic representations across subjects.
>
> | Method (multi-subject) | PixCorr$\uparrow$ | SSIM$\uparrow$ | AlexNet(2) (%) $\uparrow$ | AlexNet(5) (%) $\uparrow$ | InceptionV3 (%) $\uparrow$ | CLIP (%) $\uparrow$ | Efficient Net$\downarrow$ | SwAV$\downarrow$ |
> |-----------|-----------|--------|--------------|--------------|---------------|--------|-----------------|--------|
> |MindEye2  |0.374 |0.439|97.8|99.1|96.1|93.5|0.609|0.338|
> |MindHier (Multi Subj) | 0.193     | 0.359  | 91.0        | 97.7        | 95.7         | 95.8  | 0.619           | 0.341  |
>
> ### Q2: Can the authors provide the results supporting the claim that blurred images improve low-level metrics (lines 329–330)?
>
> We appreciate the opportunity to clarify this point. This claim is substantiated by the results in Table 1, which compares our standard model MindHier against a version incorporating the auxiliary blurred image prior (MindHier$^†$).
>
> - MindHier: Pure scale-wise reconstruction achieves an SSIM of 0.381.
> - MindHier$^†$: Incorporating the auxiliary low-level feature (blurred image), similar to the strategy in MindEye2, increases the SSIM to 0.461.
>
> This comparison confirms that explicitly injecting low-level blurred priors significantly boosts structural metrics. We have updated the manuscript to explicitly reference this comparison in relation to the reviewer's query.
>
> ### Q3: Would a VQA task be an appropriate metric for evaluating faithful reconstruction?
>
> We appreciate this insightful suggestion. We agree that a Visual Question Answering (VQA) task serves as a robust metric for assessing faithful reconstruction.
>
> Following the reviewer's recommendation, we conduct a VQA evaluation using the BLIP-2-OPT-2.7B on the test set (1,000 images). As shown below, MindHier achieves higher accuracy than the current state-of-the-art (MindEye2), indicating superior preservation of faithful content.
>
> VQA Accuracy (Standard VQA metric):
> | Data Source | Accuracy |
> | :--- | :--- |
> | Ground Truth Images (Upper Bound) | 58.68% |
> | MindHier (Ours) | 43.29% |
> | MindEye2 | 41.71% |
>
> Note: An answer is considered 100% correct if it matches at least 3 annotators.
>
> We view VQA as complementary to object detection metrics, as it captures broader semantic consistency. We have added these results and a detailed discussion to **Appendix G**, and we thank the reviewer for helping us strengthen our evaluation suite.
>
> ### Q4: MindTuner is omitted from the experimental comparisons?
>
> We recognize that MindTuner is a significant baseline and have incorporated it into the revised Table 1.
>
> As shown below, MindHier outperforms MindTuner across all high-level semantic metrics (InceptionV3, CLIP, EfficientNet, and SwAV). This validates that MindHier offers superior capabilities in decoding abstract semantic content.
>
> | Method | PixCorr $\uparrow$ | SSIM $\uparrow$ | AlexNet(2) $\uparrow$ | AlexNet(5) $\uparrow$ | InceptionV3 $\uparrow$ | CLIP $\uparrow$ | Efficient Net $\downarrow$ | SwAV $\downarrow$ |
> |-----------|-----------|--------|--------------|--------------|---------------|--------|-----------------|--------|
> | MindTuner| 0.322     | 0.421  | 95.8%        | 98.8%        | 95.6%         | 93.8%  | 0.612           | 0.340  |   |
> |MindHier| 0.235     | 0.381  | 94.0%        | 98.8%        | 95.9%         | 96.4%  | 0.606           | 0.329  |

---

### Author Response · Authors · 2025-11-26

**We would like to thank the reviewers for their careful and constructive comments.** We also thank the reviewers for acknowledging our work is well-motivated and technically sound (Reviewer Pfeq, mNCs, YmRz), novel, innovative, or a promising direction (Reviewer Pfeq, u1wj, DrWf), well-written and easy to follow (Reviewer Pfeq, DrWf, YmRz).

The paper has been revised in accordance with the reviewers’ comments and suggestions. Updates and changes are marked by blue color in the revised version. We respond to the reviewers' questions point-by-point in the corresponding comments below.

Please let us know if you need further information. We look forward to hearing from you.

---

### Author Response · Authors · 2025-11-30
**Rebuttal Summary**

We express our sincere gratitude to the Area Chair for their hard work regarding the unexpected events, and to the reviewers for their constructive comments. We believe that the substantial new experimental evidence, particularly the new dataset and a new VQA evaluation, comprehensively addresses the reviewers' key concerns. These suggestions and updates significantly strengthen the manuscript and provide a solid foundation for a more positive assessment.

We have gone beyond simple clarifications, conducting **resource-intensive experiments** to empirically substantiate every claim. Below is a summary of how we have addressed the key issues:

**1. Major New Experiments: Generalizability on New Dataset**
To address concerns regarding dataset dependence (Reviewer **DrWf**), we successfully extend our training/evaluation to the **THINGS-fMRI dataset**. Using a rigorous experimental setup identical to the recent SOTA *BrainFlora (ACM MM 2025)*, MindHier achieves superior performance (e.g., **+19.7%** in AlexNet(5) and **+15.5%** in CLIP accuracy). This strongly validates that our hierarchical guidance generalizes robustly across different data distributions and acquisition protocols.

**2. Validated Cross-Subject Transferability**
Responding to requests from Reviewers **YmRz** and **Pfeq**, we implemente a multi-subject pretraining protocol. MindHier achieves a CLIP score of **95.8%** in this setting, surpassing the strong MindEye2 baseline (93.5%). This confirms that our architecture learns robust, transferable neural representations.

**3. Strengthened Baselines & State-of-the-Art Confirmation**
We address the "missing baselines" concern (Reviewers **YmRz**, **mNCs**) by adding comparisons to **Neuropictor** and **MindTuner** in Table 1. MindHier outperforms both on high-level semantic metrics while maintaining a inference speed advantage. We have also clarified the distinction between our *dynamic, scale-aware guidance* and the *static conditioning* used in prior works like Neuropictor.

**4. New Metrics: VQA, Stability, and Determinism**
*   **Faithfulness:** At the suggestion of Reviewer **YmRz**, we introduce a **Visual Question Answering (VQA)** evaluation. MindHier outperforms MindEye2 (43.29% vs 41.71%), proving it better preserves semantic content interpretable by VLMs.
*   **Stability:** Addressing Reviewer **3r2V** and **mNCs**, we quantifie generation stability. MindHier achieves significantly higher inter-trial consistency (SSIM 0.5338) compared to stochastic diffusion baselines (~0.45).
*   **Determinism:** For Reviewer **u1wj**, we provide **Single-shot (N=1)** results. MindHier could achieve SOTA-level performance in **sub-second time (0.92s)** without needing the "Best-of-N" selection required by other methods.

**5. Deep Dive into Efficiency: A Structural Paradigm Shift**
Addressing Reviewers **DrWf** and **YmRz**, we provided a granular breakdown of inference mechanics to prove our speedup is structural, not merely distinct engineering.
*   **Stage-wise Breakdown:** We showed that coarse semantic planning (Stages 0–4) is negligible (<0.1s), with computation efficiently back-loaded to high-resolution refinement.
*   **Structural Advantage:** Unlike diffusion models that iterate repeatedly at full resolution ($N_{steps} \times Cost_{FullRes}$), MindHier eliminates the refinement bottleneck. We demonstrated that forcibly reducing diffusion steps to match our speed results in catastrophic quality loss, confirming our efficiency is an intrinsic architectural advantage.

**6. Comprehensive Qualitative Transparency**
To address Reviewer **DrWf’s** request for more transparency, we added a massive gallery of **90 reconstructed images** (Appendix H) across four subjects, including failure cases. This provides an unbiased view of the model's capabilities.

**Conclusion**
In addition to the extensive experimental updates detailed above, specifically the **THINGS dataset validation, multi-subject experiments, and VQA evaluation**, we have meticulously addressed technical ambiguities regarding ROI definitions and hyperparameter specifications. Furthermore, we have conducted a thorough review of the manuscript to fix typos and enhance clarity throughout the text. We believe these comprehensive revisions solidify the manuscript as a rigorous, cognitively grounded contribution to fMRI-to-image reconstruction, and we trust that it now meets the high standards for acceptance.

---

### Meta-Review · Area_Chair_2ev6 · 2026-01-06

**Summary:**

The paper presents a well-motivated and technically sound alternative to diffusion-based fMRI-to-image reconstruction, introducing hierarchical alignment and scale-aware autoregressive generation to overcome limitations of static guidance in prior diffusion approaches.

The initial reviews were mixed (6,4,4,6,4,2), but the authors made a substantial effort to address the concerns with extensive new experiments and clarifications. In particular, they directly tackled the critique about reliance on low-resolution settings, added missing baselines such as MindTuner, and expanded evaluation to an additional dataset (THINGS) as requested. The revision also strengthens the empirical evidence with extensive cross-subject validation, broader qualitative and quantitative comparisons, and additional ablations that clarify the contribution of each component. Overall, the updated submission resolves most of the major issues raised during review, and the resulting paper now provides a clearer and more convincing case for its contributions, so it appears ready for ICLR publication.

**Reviewer Concerns:**

.

**Reviewer Scores:**

.

---

### Decision · Program_Chairs · 2026-01-26

Accept (Poster)